# Effects of Resistance Training Performed with Different Loads in Untrained and Trained Male Adult Individuals on Maximal Strength and Muscle Hypertrophy: A Systematic Review

**DOI:** 10.3390/ijerph182111237

**Published:** 2021-10-26

**Authors:** Marcio Lacio, João Guilherme Vieira, Robert Trybulski, Yuri Campos, Derick Santana, José Elias Filho, Jefferson Novaes, Jeferson Vianna, Michal Wilk

**Affiliations:** 1Postgraduate Program in Physical Education, Federal University of Juiz de Fora (UFJF), Juiz de Fora 36036-900, Brazil; marciolacio@gmail.com (M.L.); reiclauy@hotmail.com (Y.C.); joseeliasfilho@yahoo.com.br (J.E.F.); jeferson.vianna@ufjf.edu.br (J.V.); 2Strength Training Research Laboratory, Federal University of Juiz de Fora (UFJF), Juiz de Fora 36036-900, Brazil; 3Laboratory of Exercise Physiology and Morphofunctional Evaluation (LABFEX), Granbery Methodist College, Juiz de Fora 36010-359, Brazil; dericksantana22@gmail.com; 4Department of Medical Sciences, The Wojciech Korfanty School of Economics, 40-065 Katowice, Poland; rtrybulski@o2.pl; 5Study Group and Research in Neuromuscular Responses, Federal University of Lavras (UFLA), Lavras 37200-900, Brazil; 6Postgraduate Program in Physical Education, Federal University of Rio de Janeiro (UFRJ), Rio de Janeiro 21941-901, Brazil; jeffsnovaes@gmail.com; 7Institute of Sport Sciences, Jerzy Kukuczka Academy of Physical Education in Katowice, 40-065 Katowice, Poland; m.wilk@awf.katowice.pl

**Keywords:** adult male, high load, low load, morphological adaptations, strength training

## Abstract

The load in resistance training is considered to be a critical variable for neuromuscular adaptations. Therefore, it is important to assess the effects of applying different loads on the development of maximal strength and muscular hypertrophy. The aim of this study was to systematically review the literature and compare the effects of resistance training that was performed with low loads versus moderate and high loads in untrained and trained healthy adult males on the development of maximal strength and muscle hypertrophy during randomized experimental designs. The Preferred Reporting Items for Systematic Reviews and Meta-Analyses guidelines (2021) were followed with the eligibility criteria defined according to participants, interventions, comparators, outcomes, and study design (PICOS): (P) healthy males between 18 and 40 years old, (I) interventions performed with low loads, (C) interventions performed with moderate or high loads, (O) development of maximal strength and muscle hypertrophy, and (S) randomized experimental studies with between- or within-subject parallel designs. The literature search strategy was performed in three electronic databases (Embase, PubMed, and Web of Science) on 22 August 2021. *Results:* Twenty-three studies with a total of 563 participants (80.6% untrained and 19.4% trained) were selected. The studies included both relative and absolute loads. All studies were classified as being moderate-to-high methodological quality, although only two studies had a score higher than six points. The main findings indicated that the load magnitude that was used during resistance training influenced the dynamic strength and isometric strength gains. In general, comparisons between the groups (i.e., low, moderate, and high loads) showed higher gains in 1RM and maximal voluntary isometric contraction when moderate and high loads were used. In contrast, regarding muscle hypertrophy, most studies showed that when resistance training was performed to muscle failure, the load used had less influence on muscle hypertrophy. The current literature shows that gains in maximal strength are more pronounced with high and moderate loads compared to low loads in healthy adult male populations. However, for muscle hypertrophy, studies indicate that a wide spectrum of loads (i.e., 30 to 90% 1RM) may be used for healthy adult male populations.

## 1. Introduction

Resistance training is an exercise intervention that is used to develop maximal strength and muscle hypertrophy [1]. Commonly, this type of exercise is used to improve athletic performance and functional capacity during the activities of daily living in healthy and clinical populations [2,3,4]. In this way, proper manipulation of variables that make up resistance training, such as muscle action, external load, number of performed repetitions, sets, rest intervals, movement tempo, type and sequence of exercises, frequency of training, and level of effort, are considered fundamental components for maximizing neural and morphological adaptations [5,6]. Moreover, the magnitude of the load, referred to as the amount of external load during one or more sets of a given exercise, is characterized as one of the most significant aspects of resistance training programs [5,7]. The American College of Sports Medicine guidelines have declared that loads ≥80% of 1 repetition maximum (RM) are required to enhance maximal strength, while loads between 70 and 85% 1RM are necessary to increase muscle hypertrophy [5]. These recommendations are based on the repetition continuum, which proposes that the number of repetitions performed at a given load will result in specific adaptations [8].

The repetition continuum is supported by some theories, one of them is the size principle proposed by Henneman et al. [9], which advocates that higher training loads must be used to recruit motor units with a higher excitability threshold and, thus, achieve optimal muscle adaptations. On the other hand, Delorme [10] proposed that high loads increase maximal strength/power while low loads improve muscular endurance. Previous studies have provided partial support for Delorme’s hypothesis, underpinning what is currently accepted as theory [11,12]. Although some guidelines suggest the use of high and moderate loads to development maximal strength and muscle hypertrophy, several studies showed increases in maximal strength and muscle hypertrophy after resistance training with low loads (i.e., <60% 1RM) [13,14,15,16,17]. These studies are in line with recent guidelines indicating that the athletic population may achieve comparable muscle hypertrophy across a wide spectrum of loading zones [18]. However, meta-analysis data provided no differences in muscle hypertrophy between high and low loads (≥80% 1RM, or ≤8RM vs. <60% 1RM, or >15RM), moderate and low loads (60–79% 1RM, or 9–15RM vs. <60% 1RM, or >15RM), or high and moderate loads (≥80% 1RM, or ≤8RM vs. 60–79% 1RM, or 9–15RM) during resistance training performed until volitional failure in healthy adults [19]. Muscle failure was shown to be an important factor for low loads to be as effective as high loads regarding muscle hypertrophy [20,21,22,23]. On the other hand, meta-analytic data confirmed that for optimal strength gains, high loads are necessary. For example, Schoenfeld et al. [24] reported a moderate-to-large effect size (ES) difference (ES = 0.58) favoring high- (>60% 1RM) vs. low- (60% 1RM) load training based on pooled data from 14 included studies.

Recently, the role of loading during resistance training aiming at skeletal muscle hypertrophy has been widely discussed [25]. Schoenfeld et al. [26] assessed the changes in maximal strength and muscle hypertrophy of the soleus (predominantly slow twitch fibers) and gastrocnemius muscles (similar composition of slow and fast twitch fibers) using low (i.e., 20–30RM) and high loads (i.e., 6–10RM) and found no additional benefit to muscle hypertrophy when resistance training was based on muscle fiber composition. The results indicated that muscle hypertrophy seems to be independent of muscle fiber type and the load used during the resistance training program, as long as the sets are performed with a high level of effort. Furthermore, a recent meta-analysis found no significant differences between low- and high-load resistance training on type I or type II muscle fiber hypertrophy [27]. Other studies also reported that 10 weeks of resistance training to volitional failure in untrained males at low and high loads (30% and 80% 1RM) resulted in similar increases in muscle fiber cross-sectional area (CSA) [23,28]. In contrast, when comparing low repetitions (3–5RM), intermediate repetitions (9–11RM), and high repetitions (20–28RM), Campos et al. [29] concluded that the three main types of muscle fibers (types I, IIA, and IIX) were increased in the low- and intermediate-repetitions groups; however, maximal strength development was greater for the low-repetitions group.

Understanding how the load can interfere on the development of maximal strength and muscle hypertrophy is critical for coaches and resistance training practitioners to be able to design more efficient training programs. However, previous systematic reviews and meta-analyses examined the development of maximal strength and muscle hypertrophy over very wide load ranges (i.e., low (≤60% 1RM) and high loads (>60% 1RM or ≥65% 1RM)) [24,30]. This division of load ranges can be a mistake for not considering the previous guidelines that divided resistance training into low (<67% 1RM or >12 repetitions), moderate (between 67 and 85% 1RM or 6–12 repetitions), and high loads (>85% 1RM or <6 repetitions) [7]. In addition, this division into a wide load range may also make it difficult to analyze the hypertrophic response and maximal strength at a well-defined load spectrum, which may extrapolate the effectiveness of the load ranges that were established and recommended in these studies and compromise the resistance training prescription. Thus, the aim of the present study was to systematically review the literature and compare the effects of resistance training performed with low loads versus moderate and high loads in untrained healthy adult males on the development of maximal strength and muscle hypertrophy during randomized experimental designs.

## 2. Materials and Methods

This systematic review followed the Preferred Reporting Items for Systematic Reviews and Meta-Analyses (PRISMA) guidelines [31,32].

### 2.1. Eligibility Criteria

Following the PRISMA guidelines, relevant studies were included according to participants, intervention, comparators, outcomes, and study design (PICOS) [33] inclusion criteria (Table 1). It is noteworthy that only studies in the English language and published in peer-reviewed journals were included. Furthermore, training status was established according to untrained: individuals who had not been consistently trained for 1 year; recreationally trained: individuals training consistently for 1–5 years; highly trained: individuals training for at least 5 years [34].

### 2.2. Information Sources

Searches were performed in three electronic databases (Embase, PubMed, and Web of Science) on 22 August 2021. The studies were retrieved from an electronic database search, comprehensive sweeping in the reference list of the included studies, and systematic reviews with meta-analyses previously published on load and chronic adaptations, more specifically on the development of maximal strength and muscle hypertrophy.

### 2.3. Search Strategy

The search strategy combined the descriptors using the Booleans operators (AND/OR/NOT) in the following way: (“resistance training” OR “strength training” OR “resistance exercise”) AND (“high-load” OR “high load” OR “high-intensity” OR “high intensity” OR “heavy loads” OR “low-load” OR “low load” OR “low-intensity” OR “low intensity” OR “volume training” OR “training load”) AND (“hypertrophy” OR “muscle size” OR “skeletal muscle enlargement” OR “muscle thickness” OR “muscle mass” OR “muscle fibers, skeletal” OR “muscle, skeletal” OR “growth” OR “cross-sectional area” OR “muscle strength” OR “dynamic strength” OR “dynamic force” OR “maximum repetition” OR “1RM” OR “isometric contraction” OR “isometric force” OR “maximal voluntary contraction” OR “MVC” OR “maximal voluntary isometric contraction” OR “MVIC”) (Appendix A).

### 2.4. Selection Process

The studies that were retrieved from each database were clustered using the EndNote X9 software (Clarivate Analytics, Philadelphia, PA, USA) and duplicate studies were automatically and manually removed. The titles and abstracts were assessed according to the eligibility criteria by two independent researchers (M.L. and D.S.). Conflicts were settled by a third reviewer (J.G.V.). The researchers were not blinded to authors, institutions, or journals. The abstracts not offering enough information to be evaluated were sent to the next phase, in which the full text was read.

### 2.5. Data Collection Process and Data Items

Two independent reviewers (M.L. and D.S.) extracted the data from the full texts. Data were recorded in Excel spreadsheets that were created specifically for this review. The data that was collected covered the characteristics of participants (sample size, age, height, body mass, and training status) and characteristics of the studies (study design, time of analysis, resistance exercise(s), prescription, weekly frequency, movement tempo, volume, and findings). After the selection, the data extracted by both reviewers were compared and the divergences were decided by all three reviewers (M.L., D.S., and J.G.V.).

### 2.6. Methodological Quality Assessment

After the literature search and selection, a methodological quality assessment was performed independently by two authors (J.E.F. and J.G.V.) using the Physiotherapy Evidence Database (PEDro) scale, which is a valid measure of the methodologic quality of randomized trials [35] and displays acceptable inter-rater reliability [35,36]. Thus, scores on the PEDro scale ranged from 0 (low methodological quality) to 10 (high methodological quality). The quality of the studies was used for qualitative assessment, and it was not used as an exclusion criteria. The methodological quality of the study was categorized as follows: a score ranging from 6 to 10 was indicative of high quality, whereas scores of 4–5 indicated moderate quality and scores ≤3 indicated low quality [37].

### 2.7. Synthesis Methods

A quantitative analysis of the data (meta-analysis) was not performed. A narrative synthesis of the results was provided.

## 3. Results

### 3.1. Study Selection

A flow diagram of the literature search is presented in Figure 1. A total of 3913 studies were generated by the database search. Afterward, 1778 duplicate studies were removed. In addition, another 2091 studies were removed after analyzing each title and abstract. Then, 54 studies were considered for the systematic review, with 10 studies considered by checking the reference list of included studies. After analyzing the eligibility criteria, 31 studies were excluded and, finally, 23 studies remained for the qualitative synthesis.

### 3.2. Study Characteristics

Table 2 shows the characteristics of the participants in the 23 studies that were selected for systematic review regarding the sample size, age, height, weight, and training status (mean ± SD) of the 563 participants, where 454 were untrained (80.6%) [11,14,15,16,17,21,22,23,26,28,29,38,39,40,41,42,43,44,45] and 109 were recreationally trained (19.4%) [13,46,47] in resistance training.

Table 3 shows the characteristics of the studies that were selected for the systematic review regarding the study design, time of analysis, resistance exercise(s), prescription, weekly frequency, movement tempo, volume, and findings. Regarding the assessment of maximal strength development, 13 studies assessed dynamic strength using 1RM (56.5%) [11,13,14,15,17,21,22,23,29,44,46,47,48] and another four studies assessed the isometric strength (17.4%) [26,38,39,45]. In addition, five studies simultaneously assessed dynamic strength using 1RM and isometric strength using a maximal voluntary isometric contraction (MVIC) (21.7%) [16,28,41,42,43], and, finally, one study assessed isometric strength using maximal isometric voluntary torque (4.4%) [40]. Eight studies assessed muscle hypertrophy using ultrasound (34.8%) [14,15,17,22,26,42,43,47], whereas four verified it using a biopsy (17.4%) and six using magnetic resonance imaging (26.1%) [16,21,38,39,41,45]. In addition, one study assessed muscle hypertrophy using a biopsy and magnetic resonance imaging (4.4%) [28]. Another four studies did not perform any muscle hypertrophy assessment (17.4%) [11,13,40,48].

Regarding maximal strength development measured by the 1RM test, 18 studies indicated a significant improvement between pre- and post-intervention using low-, moderate-, and high-load training protocols [11,13,14,15,16,17,21,22,23,26,28,29,41,43,44,46,47,48]. However, when a t-test for independent samples or ANOVA were used to compare differences between groups, most studies found that the moderate- and high-load groups significantly improved their 1RM compared to the low-load group [11,17,22,28,29,41,42,43,46,48], although eight studies did not observe differences between these groups [13,14,15,16,21,23,26,44]. Greater increases in isometric strength using the MVIC test were also observed in the moderate- and high-load groups compared to the low-load groups [16,38,39,41,42,43].

For muscle hypertrophy, 16 studies showed that the low-, moderate-, and high-load groups improved the cross-sectional area and muscle thickness between pre- and post-intervention [14,15,17,21,22,23,26,28,38,39,41,42,43,45,46,47]. Moreover, comparisons between groups (i.e., low, moderate, and high load) did not show significant differences in the cross-sectional area or muscle thickness [14,15,17,21,23,26,28,38,39,42,43,45,46,47] when the repetitions of each set were performed to muscular failure [21]. However, low loads (i.e., ≤20%) seemed to be ineffective for muscle growth [22].
ijerph-18-11237-t003_Table 3Table 3Summary and characteristics of the studies included in the review.StudyDesignDuration (Weeks)ExercisePrescriptionFrequency (Days)Movement TempoVolumeOutcomes MeasuresFindingsAnderson and Kearney [11]Between-subject9Bench press**LL 1**: 1 set 100–150RM**LL 2**: 2 sets 30–40RM**ML**: 3 sets 6–8RM2 min interval3**LL 1 and LL 2**: 40 reps/minNot equalized1RM strength (bench press)**1RM**: ↑ pre- to post-intervention in all groups (ML > LL 1 and LL 2)Campos et al. [29]Between-subject8Leg press, squat, and leg extension**LL**: 2 sets 20–28RM, 1 min interval**ML**: 3 sets 9–11RM, 2 min interval**HL**: 4 sets 3–5RM, 3 min interval2 (4 weeks)3 (4 weeks)Not reportedEqualized1RM strength (squat, leg press, and knee extension)Muscle fCSA (biopsy of VL)**1RM**: ↑ pre- to post-intervention in all groups (leg press and back squat—HL > ML and LL; leg extension—HL > LL)**CSA**: ↑ pre- to post-intervention in ML and HLFink et al. [38]Between-subject8Barbell curl, preacher curl, hammer curl, close grip bench press, french press, and dumbbell extension**LL**: 3 sets 20RM, 30 s interval**ML**: 3 sets 8RM, 3 min interval31 s concentric2 s eccentricNot equalizedMVIC strength (elbow flexors)Muscle CSA (MRI of elbow flexors)**MVIC**: ↑ pre- to post-intervention in ML**CSA**: ↑ pre- to post-intervention in all groupsFink et al. [39]Between-subject8Unilateral bicep preacher curl**LL**: 3 sets 30% 1RM to volitional failure**ML**: 3 sets 80% 1RM to volitional failure31 s concentric2 s eccentricNot equalizedMVIC strength (elbow flexors)Muscle CSA (MRI of elbow flexors)**MVIC**: ↑ pre- to post-intervention in ML**CSA**: ↑ pre- to post-intervention in all groupsFisher et al. [40]Within-subject6Unilateral leg extension**LL**: 3 sets 50% MVIT to failure,**ML**: 3 sets 80% MVIT to failure,2 min interval12 s concentric1 s isometric3 s eccentricNot equalizedMVIT strength (knee extension)**MVIT**: ↑ pre- to post-intervention in all conditionsHolm et al. [41]Within-subject12Unilateral leg extension**LL**: 10 sets 36 (15.5% 1RM)**ML**: 10 sets 8 (70% 1RM)3**LL**: 5 s per repetition**ML**: 3 s per repetitionEqualized1RM strength (knee extension)MVIC strength (knee extension)Muscle CSA (MRI of QF)**1RM**: ↑ pre- to post-intervention in all groups (ML > LL)**MVIC**: ↑ pre- to post-intervention in ML**CSA**: ↑ pre- to post-intervention in all groups (ML > LL)Jenkins et al. [42]Between-subject4Dumbbell bicep curls**LL**: 3 sets 30% 1RM to failure**ML**: 3 sets 80% 1RM to failure2 min interval31 s concentric1 s eccentricEqualized1RM strength (elbow flexion)MVIC strength (elbow flexion)Muscle thickness (ultrasound of BB and BR)**1RM**: ↑ pre-intervention to post 2 and 4 weeks in ML**MVIC**: ↑ pre-intervention to post 2 and 4 weeks in ML**MT**: ↑ pre-intervention to post 2 and 4 weeks in all groupsJenkins et al. [43]Between-subject6Leg extension**LL**: 3 sets 30% 1RM to failure**ML**: 3 sets 80% 1RM to failure2 min interval31 s concentric1 s eccentricEqualized1RM strength (knee extension)MVIC strength (knee extension)Muscle thickness (ultrasound of VL, VM, and RF)**1RM**: ↑ pre- to post-intervention in all groups (ML > LL)**MVIC**: ↑ pre- to post-intervention in all groups (ML > LL)**MT**: ↑ pre- to post-intervention in all groupsJones et al. [48]Between-subject10Squat (olympic style), Romanian dead lift, lunge, half back squat**LL**: 4 sets 5–15 (40–60% 1RM)**ML**: 4 sets 3–10 (70–90% 1RM)Both groups performed 3 sets with full range of motion and one set with partial range of motion2Not reportedNot equalized1RM strength (squat)**1RM**: ↑ pre- to post-intervention in all groups (ML > LL)Lasevicius et al. [22]Within-subject12Unilateral leg press and unilateral bicep curl**LL (20%)**: 3 sets at 20% 1RM to concentric muscle failure**LL (40%)**: 4 sets at 40% 1RM to volitional failure**LL**: 5 sets at 60% 1RM to volitional failure**ML**: 4 sets at 80% 1RM to volitional failure2 min interval22 s concentric2 s eccentricEqualized1RM strength (leg press and elbow flexion)Muscle thickness (ultrasound of elbow flexors and VL)**1RM bicep curl**: ↑ pre- to post-intervention in all conditions (80% > 20, 40, and 60%) **1RM leg press**: ↑ pre- to post-intervention in all conditions (60 and 80% > 20 and 40%)**CSA**: ↑ pre- to post-intervention in all conditions (80, 60, and 40% > 20%)Lasevicius et al. [21]Within-subject8Unilateral leg extension**LL to failure**: 3 sets at 30% 1RM**LL not to failure**: 3 sets at 30% 1RM**ML to failure**: 3 sets at 80% 1RM**ML not to failure**: 3 sets at 80% 1RM2 min interval2Not reportedEqualized1RM strength (knee extension)Muscle CSA (MRI of QF)**1RM**: ↑ pre- to post-intervention in all conditions (ML failure = ML not to failure > LL failure = LL not to failure)**CSA**: ↑ pre- to post-intervention in ML failure, ML not to failure, and LL failureLim et al. [23]Between-subject10Leg press, leg extension, and leg curl**LL**: 3 sets 30% 1RM to volitional failure**LL WM**: 3 sets 30% 1RM (work matched to HL)**ML**: 3 sets 80% 1RM to volitional failureInterval not reported3Not reportedEqualized to LL WM and ML1RM strength (leg extension)Muscle fCSA (biopsy of VL)**1RM**: ↑ pre- to post-intervention in all groups**CSA**: ↑ pre- to post-intervention in type I in LL and ML; ↑ pre- to post-intervention in type II in all groupsLopes et al. [13]Between-subject6**Session A**: bench press (flat and inclined), fly, bicep curls (standing and concentrated), squat, leg press, and abdominal;**Session B**: back lat pull-down, one arm dumbbell row, row (open grip), tricep overhead extension, tricep pushdown, leg curl, calf raise, and abdominal**LL**: 3 sets 20RM**ML**: 6 sets 10RM1 min interval4Not reportedEqualized1RM strength (bench press and squat)**1RM**: ↑ pre- to post-intervention in bench press and back squat in all groupsMitchell et al. [28]Within-subject10Unilateral leg extension**LL**: 3 sets 30% 1RM to point of fatigue**ML**: 1 set 80% 1RM to voluntary failure**ML**: 3 sets 80% 1RM to point of fatigue1 min interval3Not reportedNot equalized1RM strength (knee extension)MVIC strength (knee extensors)Muscle CSA (MRI of QF)Muscle fCSA (biopsy of VL)**1RM**: ↑ pre- to post-intervention in all conditions (ML 1 and ML 3 > LL)**CSA**: ↑ pre- to post-intervention in all groupsMorton et al. [46]Between-subject12**Monday/Thursday**: leg press/seated row, barbell bench press/cable hamstring curl, and front planks; **Tuesday/Friday**: machine-guided shoulder press/bicep curls, tricep extension/wide-grip pulldown, and machine-guided leg extension**LL**: 3 sets 20–25 (30–50% 1RM) volitional failure**ML**: 3 sets 8–12 (75–90% 1RM) volitional failure1 min interval4Not reportedNot equalized1RM strength (bench press, leg press, shoulder press, and knee extension)Muscle fCSA (biopsy of VL)**1RM**: ↑ pre- to post-intervention in all groups (ML > LL in bench press)**CSA**: ↑ pre- to post-intervention in all groupsNetreba et al. [44]Between-subject8Leg press**LL**: 7 sets 35–50 (20–25% 1RM), 5–6 min interval**ML**: 5 sets (2 × 40–50 s (60–70% 1RM), 5–6 min interval**HL**: 6–10 reps (85–90% 1RM), 5–6 min interval3**LL**: high velocity**ML**: low velocity**HL**: average velocityEqualized1RM strength (knee extension)Muscle fCSA (biopsy of VL)**1RM**: ↑ pre- to post-intervention in LL and HL in all velocities; ↑ pre- to post-intervention in ML for 180°/s, 300°/s, and isometric strength**CSA**: ↑ pre- to post-intervention in type I and II in all groups, except type II of the LL groupsNóbrega et al. [14]Within-subject12Unilateral leg extension**LL to failure and LL to volitional interruption**: 3 sets 30% 1RM**ML to failure and ML to volitional interruption**: 3 sets 80% 1RM2 min interval2Not reportedNot equalized1RM strength (knee extension)Muscle CSA (ultrasound of VL)**1RM**: ↑ pre- to post-intervention in all groups**CSA**: ↑ pre- to post-intervention in all groupsPopov et al. [45]Between-subject8Leg press**LL**: session 1-3 sets (4 × 50–60 s 50% MVIC) + session 2 and 3-1 set (4 × 50–60 s), 10 min interval**ML**: session 1-7 sets 6–12 (80% MVIC) 10 min interval + session 2 and 3-3 sets 6-12 (80% MVIC), 10 min interval3Not reportedNot equalizedMVIC strength (knee extensors)Muscle CSA (MRI of QF and GM)**MVIC**: ↑ pre- to post-intervention in all groups**CSA**: ↑ pre- to post-intervention in all groupsSchoenfeld et al. [47]Between-subject8Flat barbell press, barbell military press, wide-grip lat pulldown, seated cable row, squat, leg press, and leg extension**LL**: 3 sets 25–35 (30–50% 1RM) to failure**ML**: 3 sets 8–12 (70–80% 1RM) to failure90 s interval3 days per week1 s concentric2 s eccentricNot equalized1RM strength (squat and bench press)Muscle thickness (ultrasound of elbow flexors, elbow extensors, and QF)**1RM**: ↑ pre- to post-intervention in all groups, except in bench press in LL**CSA**: ↑ pre- to post-intervention in all groupsSchoenfeld et al. [26]Within-subject8Seated plantar flexion and standing plantar flexion**LL**: 4 sets 20–30RM**ML**: 4 sets 6–10RM90 s interval2Controlled concentric~2 s eccentricNot equalizedMVIC strength (plantar flexors)Muscle thickness (ultrasound of medial gastrocnemius, lateral gastrocnemius, and soleus)**MVIC**: ↑ pre- to post-intervention in all groups**MT**: ↑ pre- to post-intervention in all groupsTanimoto and Ishii [16]Between-subject12Leg extension**LL 1**: 3 sets 50% 1RM to failure**LL 2**: 3 sets 50% 1RM to failure**ML**: 3 sets 80% 1RM to failure1 min interval3 days per week**LL 1**: 3 s eccentric and 3 s concentric with 1 s pause with no relaxation**LL 2 and ML**: 1 s eccentric and 1 s concentric with 1 s pauseNot equalized1RM strength (knee extension)MVIC strength (knee extensors)Muscle CSA (MRI of QF)**1RM**: ↑ pre- to post-intervention in all groups**MVIC**: ↑ pre- to post-intervention in ML**CSA**: ↑ pre- to post-intervention in MLTanimoto et al. [17]Between-subject13Squat, bench press, lat pulldown, abdominal bend, and back extension**LL**: 3 sets ~ 55–60% 1RM to failure**ML**: 3 sets ~ 80–90% 1RM to failure1 min interval2**LL**: 3 s concentric3 s eccentric**HL**: 1 s concentric1 s eccentric1 s restNot equalized1RM strength (squat, chest press, lat pull- down, ab bend, back extension, and knee extension)Muscle thickness (ultrasound of chest, anterior upper arm, posterior upper arm, abdomen, subscapula, anterior thigh, and posterior thigh)**1RM**: ↑ pre- to post-intervention in all groups (back extension ML > LL)**MT**: ↑ pre- to post-intervention in all groupsTanimoto et al. [15]Between-subject13Squat**LL**: 3 sets ~ 55–60% 1RM to failure**ML**: 3 sets ~ 85–90% 1RM to failure1 min interval2**LL**: 3 s concentric3s eccentric**HL**: 1 s concentric1 s eccentric1 s restNot equalized1RM strength (squat)Muscle thickness (ultrasound of anterior thigh and posterior thigh)**1RM**: ↑ pre- to post-intervention in all groups**MT**: ↑ pre- to post-intervention in all groupsBB: biceps brachii; BR: brachialis; CSA: cross-sectional area; fCSA: fiber cross-sectional area; GM: gluteus maximus; HL: high load; LL: low load; min: minutes; ML: moderate load; MRI: magnetic resonance imaging; MT: muscle thickness; MVIC: maximum isometric voluntary contraction; MVIT: maximal voluntary isometric torque; QF: quadriceps femoris; RF: rectus femoris; RM: repetition maximum; s: seconds; VL: vastus lateralis; VM: vastus medialis; WM: work matched; ↑: denotes significant increases; ↓: denotes significant decreases; >: denotes significant difference between groups; ~: approximately.


### 3.3. Methodological Quality in the Included Studies

Table 4 shows the average quality rating of studies that were evaluated using the PEDro scale. The average of 5.7 ± 0.9 indicated that the set of studies that were selected for this systematic review had moderate quality; furthermore, none of them were considered low quality.

## 4. Discussion

This study aimed to systematically review the literature and compare the effects of resistance training that was performed with low loads versus moderate and high loads in untrained healthy adult males on the development of maximal strength and muscle hypertrophy during randomized experimental designs. The main findings indicated that the load magnitude used during resistance training influences the development of maximal dynamic and isometric strength. In general, comparisons between groups (i.e., low, moderate, and high loads) showed higher gains in 1RM and MVIC when moderate and high loads were used. On the other hand, regarding muscle hypertrophy, most studies showed that when resistance training was undertaken to muscle failure, the load used seemed to have a lesser influence on muscle hypertrophy. We emphasize that all studies were classified as being of moderate to high methodological quality, although only two studies scored higher than six points.

### 4.1. Effects of Different Loads on Maximal Strength Development

Several studies suggested that performing resistance training with a high load is necessary to maximize gains in 1RM strength [11,17,21,22,28,29,42,43,46,47], although low loads were also shown to be effective in increasing these gains [11,13,14,15,16,17,22,23,44,46,47]. In this regard, a recent systematic review with a meta-analysis proposed by Schoenfeld et al. [24] showed that resistance training performed with high and low loads translated into percentage gains with 1RM of 35.4 and 28.0%, respectively. In contrast, Lasevicius et al. [22] observed a plateau in maximal strength development after 6 weeks of resistance training with low loads. Moreover, after 12 weeks, the groups that performed resistance training with high loads showed significant improvements in maximal strength compared to low load groups. As most studies assessed untrained individuals [11,14,15,16,17,21,22,28,29,42,43,44], it is possible to suggest that the early phases of resistance training were primarily impacted by enhancements in motor learning and coordination [49]. Therefore, low load training schemes might provide a sufficient stimulus to increase maximal strength. Untrained individuals generally have a lower coordination level for performing resistance exercises [22]. Thus, it seems that resistance training with low loads may be adequate to generate neural adaptations [50] and allow muscles to be controlled more effectively within the context of the task [51], thereby increasing maximal strength. However, it should be noted that the ability to produce force is a result of a combination of neural factors [52], muscle mass [53], and the specificity of the load during resistance training (i.e., high loads) [22]. Consequently, it is undeniable that heavier load training may be increasingly important to achieve maximal strength gains as the individual becomes more experienced in resistance training [24,30]. In this line, high loads may be more effective at increasing the recruitment of motor units and promote changes in agonist-antagonist co-activation ratios in the long term compared to low loads [54].

Other studies also showed that heavier loads were superior to lighter loads at improving MVIC [38,39,42,43]. In this way, Schoenfeld et al. [26] found no difference between low and moderate loads on MVIC. Thus, it seems that high loads may lead to greater increases in neural drive compared to low loads [54,55], improving performance in this specific situation. Despite this, few studies assessed the effect of resistance training that was performed to muscle failure and not performed to muscle failure, with the objective of maximal strength development [21,22]. There is a consensus that muscle failure is not necessary to maximize strength development in untrained males [14,21]. Thus, it is possible to infer that muscle failure is not a key factor for increasing maximal strength, although studies with trained male individuals are still needed to confirm this hypothesis.

### 4.2. Effects of Different Loads on Muscle Hypertrophy Development

Most of the studies included in this review showed no significant difference in muscle hypertrophy and muscle thickness when low, moderate, and high loads were compared [14,15,17,21,22,23,28,29,38,42,43,44,45,46,47]. Although these results are in agreement with the systematic reviews and meta-analyses that were previously published [19,24,30], we highlight that few experimental studies used the biopsy technique for a direct histological assessment of muscle hypertrophy [23,28,29,44,46]. In this way, conclusions based on the current literature can only be extrapolated to imaging techniques that have assessed the muscle as a whole (i.e., ultrasound and magnetic resonance imaging). Indeed, there is contradictory evidence regarding the agreement between imaging and histological techniques regarding muscle hypertrophy determination [56] in which it is not possible to state whether type I and II fibers respond similarly to resistance training schemes (i.e., low and high loads) [57]. In untrained male individuals, Campos et al. [29] observed that after 6 weeks of resistance training for lower limbs, the low-load group did not show a significant increase in type I fibers, while the high-load group demonstrated a significant increase in type I, IIa, and mainly IIX fibers. In contrast, Mitchell et al. [28] and Lim et al. [23] verified a significant increase for all types of muscle fibers for both resistance training schemes (i.e., low and high load). To assess trained male individuals, Morton et al. [46] randomized 49 young males with an average experience of 4 years in resistance training for 12 weeks of training sessions for the whole body using low- and high-load schemes. The results of this study showed that both load schemes yielded similar rates of muscle hypertrophy in all types of fibers that were assessed. In isolation, these results suggest that individuals may present similar growth in all types of muscle fibers regardless of the load scheme used. However, it is currently impossible to determine whether the hypertrophic potential of each type of fiber for a given load scheme may increase or decrease over time [57] since most studies that investigated this topic were short in duration [23,28,46]. It is important to highlight that the muscle biopsy technique also had significant limitations, such as the extraction of a small portion of the skeletal muscle that may not necessarily reflect the muscle hypertrophy of the analyzed muscle as a whole [57]. Additionally, studies that applied this technique chose the vastus lateralis for the analysis [23,28,29,46], which limits the extrapolation of results to other muscles that might respond differently to the same exercise protocol. Another important point that is currently discussed is whether muscle hypertrophy may be enhanced by applying lighter loads on muscles that are predominantly composed of type I fibers and heavier loads on muscles that are predominantly composed of type II fibers. Although Schoenfeld et al. [26] verified similar muscle hypertrophy in the soleus (predominance of type I fibers) and gastrocnemius (similar composition of type I and II fibers) to both load schemes (i.e., heavy and light) after 8 weeks of resistance training in untrained young subjects, factors such as the assessed muscles, the method used to assess hypertrophy (i.e., ultrasound), and training status make it difficult to extrapolate the results, meaning that further studies are required.

The results of this research also indicate that for resistance training with low loads has hypertrophic gains similar to high loads, yet the repetitions must be performed to muscular failure [14,21,22,23,28,39,42,43,45,46,47]. Recently, Lasevicius et al. [21] submitted 25 untrained males to four unilateral knee extension protocols. The high-load protocols with and without failure were performed at 80% 1RM, while the low-load protocols that were performed in the same manner were performed at 30% 1RM. The results of this study showed that when resistance training was performed with a high load, muscle failure did not seem to be a fundamental factor of achieving muscle hypertrophy. On the other hand, when the training protocol was performed to failure with a low load, it led to greater hypertrophy (7.8%) than the one that was not performed to failure (2.8%). According to the size principle proposed by Henneman et al. [9], motor units are recruited from the smallest to largest ones (i.e., as the strength demands increase, motor units with a greater excitability threshold are added to the motor units with smaller excitability threshold to increase force). In this regard, when resistance training with low loads is conducted to muscle failure, as the repetitions are performed, fatigue increases the effort required to produce the same force [57]. In this way, motor unit recruitment is increased until the entire motor unit pool is fully activated. This may be why resistance training with high and low loads leads to similar muscular hypertrophy. Furthermore, it is conceivable that resistance training with low loads that are performed to muscle failure would result in a long time under tension [25,58], which would cause greater metabolic stress [57], in addition to the slow speed of movement caused by fatigue, which could increase the mechanical tension. This mechanism would promote changes in metabolites that have anabolic signaling proper ties (e.g., lactate, α-ketoglutarate, and phosphatidic acid), which could potentiate muscle hypertrophy in the long term [59].

Although a wide spectrum of loads can be used during resistance training when aiming at muscle hypertrophy, there seems to be a minimal load threshold for resistance training to have hypertrophic responses that are similar to training with high loads. In this line, Lasevicius et al. [22] found that when low and high loads of resistance training were performed with matched volumes, the load schemes at 40, 60, and 80% 1RM were effective at increasing muscle hypertrophy without differences between them. However, 20% 1RM proved to be a suboptimal load to induce muscle hypertrophy compared to the other loads. There is probably a minimal load threshold (% 1RM) below which the applied mechanical tension may be insufficient to mediate the hypertrophic gains; nevertheless, this threshold has not been established [22]. To date, according to the studies that were selected for this systematic review, it is possible to infer that resistance training protocols should be designed to maximize metabolic stress (i.e., use of low and moderate loads), as well as the mechanical tension (i.e., use of high loads) to reach the entire load spectrum that is directed at muscle hypertrophy. Indeed, there is no doubt that a certain degree of mechanical tension may be achieved with low loads through mechanisms such as progressive fatigue-induced increase in motor unit recruitment, as well as decreases in movement speed that increase tension on individual sarcomeres. Nevertheless, the optimal load percentage (% 1RM) that is required for low-load resistance training to induce mechanical tension comparable to high loads has not yet been established. Perhaps the best way to achieve significant metabolic stress and mechanical tension simultaneously is to use moderate loads (i.e., between 67 and 85% 1RM or 6–12 repetitions) performed to muscular failure at some point in the season; however, this should be investigated in future studies.

## 5. Conclusions

Regarding maximal strength development, the main results of this systematic review suggest that high loads promote greater improvements compared to low loads for healthy adult males. These results are probably due to the specificity transfer of the load, as well as the 1RM test that is used to evaluate the individuals. When specific factors are eliminated and an isometric test is used, these differences are attenuated. However, as individuals become experienced in resistance training, higher training loads must be prioritized.

For muscle hypertrophy, studies indicate that a wide spectrum of loads (i.e., 30 to 90% 1RM) may be used for untrained male adult individuals. Although the same trend was observed in trained healthy adult males, the limited number of studies make it difficult to extrapolate the results to this population. Furthermore, muscle failure seems to be an important component for hypertrophic gains to be achieved. Nevertheless, coaches should be aware of the fact that resistance training that is performed with low loads until muscle failure might be quite demanding for joints and tendons, possibly increasing the risks of overtraining. On the other hand, training with high loads may reduce the total training volume and hinder hypertrophic gains. Thus, it seems that alternating periods with low and high loads, in addition to moderate loads are a good strategy to ensure the continuity of adaptive processes. Furthermore, lowloads (i.e., <30% 1RM) may be inefficient to improve muscle hypertrophy, even if they are driven to muscle failure. Therefore, we suggest that future studies compare training loads below 30% 1RM with moderate and high loads to clarify the training threshold at which low loads become ineffective for muscle hypertrophy.

## Figures and Tables

**Figure 1 ijerph-18-11237-f001:**
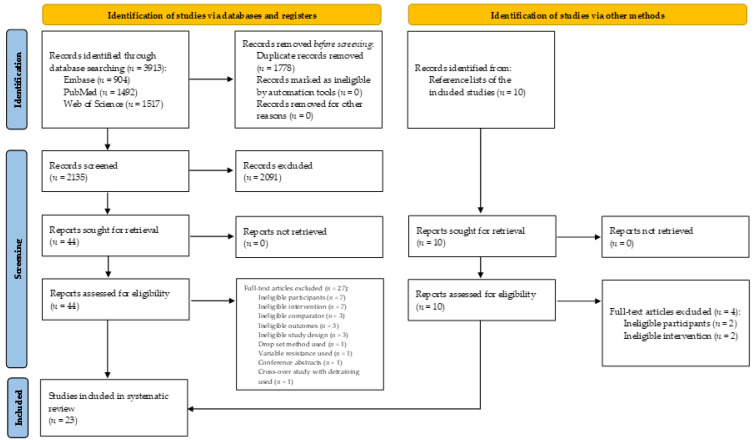
PRISMA 2020 flow diagram for this systematic review, which included searches of databases, registers, and other sources.

**Table 1 ijerph-18-11237-t001:** PICOS eligibility criteria for the inclusion of studies in the systematic review.

Parameters	Inclusion Criteria
Participants	Healthy males aged between 18 and 40 years; resistance trained or untrained; without a history of bone, muscle, or articular injury.
Intervention	Intervention performed with low loads (<67% 1RM or >12 repetitions).
Comparators	Intervention performed with moderate loads (67–85% 1RM or 6–12 repetitions) or high loads (>85% 1RM or <6 repetitions).
Outcomes	Development of maximal strength (dynamic strength or isometric strength) and muscle hypertrophy (cross-sectional area, muscle thickness, or skeletal muscle fiber size).
Study design	Experimental randomized studies with between- or within-subject parallel designs.

**Table 2 ijerph-18-11237-t002:** Characteristics of the participants.

Study	Participants (*n* = 563)	Age (years)	Height (cm)	Weight (kg)	Training Status
Anderson and Kearney [11]—mean ± SD	43	20.7 ± 1.8	180.0 ± 13.0	75.1 ± 3.9	UT
Campos et al. [29]—mean ± SD	32	22.5 ± 5.8	178.3 ± 7.2	77.8 ± 11.9	UT
Fink et al. [38]—mean ± SD	20	19.8 ± 1.0	169.3 ± 4.4	64.1 ± 7.9	UT
Fink et al. [39]—mean ± SD	21	23.2 ± 2.7	168.6 ± 4.5	63.4 ± 4.7	UT
Fisher et al. [40]—mean ± SD	7	20.6 ± 0.5	178.9 ± 3.2	77.1 ± 2.7	UT
Holm et al. [41]—mean ± SE	11	24.7 ± 1.1	183.0 ± 2.0	79.7 ± 4.0	UT
Jenkins et al. [42]—mean ± SD	15	21.7 ± 2.4	181.6 ± 7.5	84.7 ± 23.5	UT
Jenkins et al. [43]—mean ± SD	26	23.1 ± 4.7	180.6 ± 6.0	80.0 ± 14.1	UT
Jones et al. [48]—mean ± SD	26	20.1 ± 1.4	183.1 ± 5.3	80.5 ± 11.1	RT
Lasevicius et al. [22]—mean ± SD	30	24.5 ± 2.4	180.0 ± 0.7	77.0 ± 16.5	UT
Lasevicius et al. [21]—mean ± SD	25	24.0 ± 4.8	176.0 ± 6.5	74.3 ± 12.6	UT
Lim et al. [23]—mean ± SD	21	23.4 ± 1.7	174.3 ± 6.5	76.7 ± 11.1	UT
Lopes et al. [13]—mean ± SD	16	26.6 ± 6.1	176.9 ± 6.2	81.5 ± 12.6	RT
Mitchell et al. [28]—mean ± SE	18	21.0 ± 0.8	176.0 ± 4.0	73.3 ± 1.4	UT
Morton et al. [46]—mean ± SE	49	23.0 ± 1.0	181.0 ± 1.0	86.0 ± 2.0	RT
Netreba et al. [44]—mean ± SE	30	24.8 ± 3.1	178.9 ± 6.4	74.4 ± 7.3	UT
Nóbrega et al. [14]—mean ± SD	32 (5 dropped out)	23.0 ± 3.6	176.0 ± 0.6	Not reported	UT
Popov et al. [45]—mean ± SE	18	21.0 ± 2.0	181.0 ± 10.0	75.0 ± 3.0	UT
Schoenfeld et al. [47]—mean	24 (6 dropped out)	23.3	175.0	82.5	RT
Schoenfeld et al. [26]—mean	30 (4 dropped out)	22.5	175.7	77.3	UT
Tanimoto and Ishii [16]—mean ± SD	24	19.4 ± 0.7	170.3 ± 4.8	59.4 ± 5.9	UT
Tanimoto et al. [17]—mean ± SD	36	19.4 ± 0.7	174.4 ± 5.6	63.5 ± 4.2	UT
Tanimoto et al. [15]—mean ± SD	24	20.1 ± 1.1	175.1 ± 5.7	62.7 ± 4.1	UT

n: sample size; RT: recreationally trained; SD: standard deviation; SE: standard error; UT: untrained.

**Table 4 ijerph-18-11237-t004:** Methodological quality of the included studies.

Study	1	2	3	4	5	6	7	8	9	10	11	Total	Overall Quality
Anderson and Kearney [11]	Y	Y	N	Y	N	N	N	Y	Y	Y	Y	6	High
Campos et al. [29]	Y	Y	N	Y	N	N	N	Y	Y	Y	N	5	Moderate
Fink et al. [38]	Y	Y	N	Y	N	N	N	Y	Y	Y	Y	6	High
Fink et al. [39]	Y	Y	N	Y	N	N	N	Y	N	Y	Y	6	High
Fisher et al. [40]	Y	N	N	Y	N	N	N	N	Y	Y	Y	5	Moderate
Holm et al. [41]	Y	Y	N	Y	N	N	Y	Y	Y	Y	Y	7	High
Jenkins et al. [42]	Y	Y	N	Y	N	N	N	N	N	Y	Y	4	Moderate
Jenkins et al. [43]	N	Y	N	Y	N	N	N	Y	Y	Y	Y	6	High
Jones et al. [48]	Y	Y	N	Y	N	N	N	Y	Y	Y	Y	6	High
Lasevicius et al. [22]	Y	Y	N	Y	N	N	N	Y	Y	Y	Y	6	High
Lasevicius et al. [21]	Y	Y	N	Y	N	N	N	Y	Y	Y	Y	6	High
Lim et al. [23]	Y	Y	N	Y	N	N	N	Y	Y	Y	Y	6	High
Lopes et al. [13]	Y	Y	N	Y	N	N	N	Y	Y	Y	N	6	High
Mitchell et al. [28]	Y	Y	N	Y	N	N	N	Y	Y	Y	Y	6	High
Morton et al. [46]	Y	Y	N	Y	N	N	Y	N	N	Y	Y	5	Moderate
Netreba et al. [44]	N	Y	N	N	N	N	N	Y	Y	Y	Y	5	Moderate
Nóbrega et al. [14]	N	Y	N	Y	N	N	N	N	N	Y	Y	4	Moderate
Popov et al. [45]	N	N	N	Y	N	N	N	Y	Y	Y	Y	5	Moderate
Schoenfeld et al. [47]	Y	Y	N	Y	N	N	N	Y	N	Y	Y	5	Moderate
Schoenfeld et al. [26]	Y	Y	N	Y	N	Y	Y	Y	Y	Y	Y	8	High
Tanimoto and Ishii [16]	Y	Y	N	Y	N	N	N	Y	Y	Y	Y	6	High
Tanimoto et al. [17]	Y	Y	N	Y	N	N	N	Y	Y	Y	Y	6	High
Tanimoto et al. [15]	Y	Y	N	Y	N	N	N	Y	Y	Y	Y	6	High
**Mean**	-	-	-	-	-	-	-	-	-	-	-	5.7 ± 0.9	Moderate
**Median**	-	-	-	-	-	-	-	-	-	-	-	6.0	-

Y: yes; N: no. PEDro scale criteria. 1: Eligibility criteria were specified. 2: Subjects were randomly allocated to groups (in a crossover study, subjects were randomly allocated an order in which treatments were received). 3: Allocation was concealed. 4: The groups were similar at baseline regarding most important prognostic indicators. 5: There was blinding of all subjects. 6: There was blinding of all therapists/researchers who administered the therapy/protocol. 7: There was blinding of all assessors who measured at least one key outcome. 8: Measures of at least one key outcome were obtained from more than 85% of the subjects that were initially allocated to groups. 9: All subjects for whom outcome measures were available received the treatment or control condition as allocated or, where this was not the case, data for at least one key outcome were analyzed using “intention to treat.” 10: The results of between-group statistical comparisons were reported for at least one key outcome. 11: The study provided both point measures and measures of variability for at least one key outcome.

## Data Availability

The data presented in this study are available on request from the corresponding author.

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
