# Peer review of "Effects of Resistance Training Performed with Different Loads in Untrained and Trained Male Adult Individuals on Maximal Strength and Muscle Hypertrophy: A Systematic Review"

_ijerph, 2021, doi:10.3390/ijerph182111237_

Round 1

Reviewer 1 Report

I thank the authors for the changes, improvements, and clarifications.

Author Response

Reviewer 1

I thank the authors for the changes, improvements, and clarifications.

Reply: We thank for your revision and also made a grammatical correction of the text with a native speaker.

Reviewer 2 Report

Thank you for allowing me to review this article again. I think the article is better and has a clearer introduction that sets the context for where the article fits within evidence base. I do have many minor comments below that should be feasible to fix before acceptance.

Abstract

Line 28: sentence syntax which is difficult to read, change to something like: "The Preferred Reporting Items for Systematic Reviews and Meta-Analyses guidelines (2021) were followed with the eligibility criteria defined according to..."

Line 32: "...with between- or within-subject parallel designs"

Line 33: Capitalise "august"

Line 44: "with a healthy adult male population"

Line 45: Consistency with the previous sentence. If the population are the same as the previous sentence. My suggestion has been "healthy adult male population", so whatever is chosen I think it should remain consistent.

Introduction

Line 59: when referring to the guidelines themselves, the word "has" needs to change to "have". When referring to the ACSM, "has" is the correct word to use.

Line 59: About the loading guidelines, I feel like you need to say one or the other. Technically, >80% 1RM also fits the >85% 1RM recommendation. I would also be specific in terms of the repetitions that are recommended. You can perform more than 6 repetitions at 80% 1RM, so is this recommendation more like a 6RM load?

Line 93: "The results indicate that muscles respond to exercise" is redundant. I would say that "The results indicate that muscle hypertrophy seem to be independent of muscle fibre type..."

Line 113: Change "loads" to "load"

Methods

Eligibility criteria: from the abstract I believe the participants are males so I'm thinking is this part of your eligibility criteria? If so, this needs to be made in the first paragraph, as you have a table explaining the paragraph. Also, with the exclusion criteria, as you are using males, pregnancy is not required to be placed in the paragraph.

Table 1: Participants: "Healthy males aged between 18 and 40 years

Outcomes: Would it be more accurate to say "or" instead of "and" when listing the muscle hypertrophy outcomes? Saying "and" would imply that you looking for studies that measure hypertrophy using all the listed methods.

Study design: "Experimental randomized studies with between- or within-subject parallel designs"

Line 138: Capitalise "august"

Methodological quality assessment: For consistency with other studies, the higher the number for the PEDro scale, the better the quality. If future researchers attempt to synthesise studies using the PEDro scale, there would be errors in the interpretation of the evidence. I've just seen your categories have the correct interpretation so making this consistent on line 175 is needed.

Results

Table 2: Could how the data is being reported in each study be represented by a symbol rather than writing the words? 

Line 200-203: The definition of training status should be presented in the methods section.

Line 227: Differences may exist but they may not be significantly different according to null-hypothesis statistical testing. I would add "significantly" before any statement when you are mentioning no difference between groups.

Discussion

Line 264: "Influences the development of maximal dynamic and isometric strength"

Line 292-293: "Consequently, it is undeniable that heavier load training may be increasingly important..."

Line 295: "...changes in in agonist-antagonist co-activation..."

Line 299-300: Does this sentence mean that high-loads may lead to increases in neural drive compared to low-loads? If so, change to "it seems that high-loads may lead to greater increases in neural drive compared to low-loads..."

Line 359-360: I don't think preferential recruitment is the right word choice for this context. The motor units with greater excitability are added in addition to the motor units with lower excitability. So the sentence needs to be made clear that the motor units are added, rather than preferentially recruited which fits Henneman's Size Principle.

Line 362-363: This sentence doesn't really make sense, but I understand what you're trying to say. This first part of the sentence needs to say that as repetitions are performed, fatigue increases the effort required to produce the same force, which therefore increases motor unit recruitment until the entire motor unit pool is activated and fatigues with the repetitions that are being performed. This is why low-loads are comparable to high-loads, the fatigue-induced increases in effort to overcome the same load requires an increase in motor unit recruitment, which therefore increases mechanical tension, the primary driver of muscular hypertrophy.

Line 366: Not only a long time-under-tension but because of the slow movement velocity caused by fatigue, the mechanical tension is actually quite high in magnitude.

Line 369-370: The evidence for muscle hypertrophy coming from metabolic stress is mixed at best. The article cited is from 2013 and the evidence has shifted from that point. I would read Henning Wackerhage's review article which discusses metabolic stress as a synergist rather than a direct mechanism (https://pubmed.ncbi.nlm.nih.gov/30335577/)

Line 373: remove "with low-load". It's implied that you are talking about low-load because you are discussing a minimal load threshold.

Line 375: "matched volume"

Line 376: Change "meantime" to "However".

Line 380: "...not yet established"

Line 381-384: From the point above, I agree that low loads should be selected to maximise metabolic stress, but doing this also increases mechanical tension. So you're inducing sufficient levels of mechanical tension either through loading (i.e. high force output) or through fatigue-induced increases in motor unit recruitment and decreases in movement velocity, thus also increasing tension on individual sarcomeres due to the force-velocity relationship. This needs to be made clear in this paragraph because it is an important finding but the muscle physiology/mechanics needs to be applied rather than continuing to use older evidence.

Conclusions

Line 387: "Regarding maximal strength development, the main results of this systematic review..."

Line 395: Consistency, healthy adult males or however you would like to phrase it. I'm don't have an affinity for a particular phrasing, as long as it's consistent.

Line 408-410: I haven't seen this point discussed in the manuscript until now. I'm not sure how powerful it is to end the article.

Author Response

Reviewer 2

Thank you for allowing me to review this article again. I think the article is better and has a clearer introduction that sets the context for where the article fits within evidence base. I do have many minor comments below that should be feasible to fix before acceptance.

Reply: We thank for your revision and we also made a grammatical correction of the text with a native speaker. All your minor comments are answered below.

Abstract

1- Line 28: sentence syntax which is difficult to read, change to something like: "The Preferred Reporting Items for Systematic Reviews and Meta-Analyses guidelines (2021) were followed with the eligibility criteria defined according to..."

Reply: Ok, we changed the sentence as suggested (line 27-28).

2- Line 32: "...with between- or within-subject parallel designs"

Reply: Ok, we changed the sentence as suggested (line 32).

3- Line 33: Capitalise "august"

Reply: Ok, we changed the sentence as suggested (line 33).

4- Line 44: "with a healthy adult male population"

Reply: Ok, we changed the sentence as suggested (line 43-44).

5- Line 45: Consistency with the previous sentence. If the population are the same as the previous sentence. My suggestion has been "healthy adult male population", so whatever is chosen I think it should remain consistent.

Reply: Ok, we have changed the sentence as suggested for better consistency (line 45).

Introduction

6- Line 59: when referring to the guidelines themselves, the word "has" needs to change to "have". When referring to the ACSM, "has" is the correct word to use.

Reply: Ok, we changed the sentence as suggested (line 58-61).

7- Line 59: About the loading guidelines, I feel like you need to say one or the other. Technically, >80% 1RM also fits the >85% 1RM recommendation. I would also be specific in terms of the repetitions that are recommended. You can perform more than 6 repetitions at 80% 1RM, so is this recommendation more like a 6RM load?

Reply: This questioning was very necessary, we noticed that the sentence was unclear. Therefore, we rewrote the sentence in a more objective way with the information given being identical to the guidelines (line 58-61).

8- Line 93: "The results indicate that muscles respond to exercise" is redundant. I would say that "The results indicate that muscle hypertrophy seem to be independent of muscle fiber type..."

Reply: Ok, we changed the sentence as suggested (line 91-92).

9- Line 113: Change "loads" to "load"

Reply: Ok, we changed the sentence as suggested (line 112).

Methods

10- Eligibility criteria: from the abstract I believe the participants are males so I'm thinking is this part of your eligibility criteria? If so, this needs to be made in the first paragraph, as you have a table explaining the paragraph. Also, with the exclusion criteria, as you are using males, pregnancy is not required to be placed in the paragraph.

Reply: Your comment was very pertinent. We restructured the paragraph for better consistency (line 124-130).

11- Table 1: Participants: "Healthy males aged between 18 and 40 years

Reply: Ok, we changed the sentence as suggested.

12- Outcomes: Would it be more accurate to say "or" instead of "and" when listing the muscle hypertrophy outcomes? Saying "and" would imply that you looking for studies that measure hypertrophy using all the listed methods.

Reply: Excellent observation, we have changed it for better consistency.

13- Study design: "Experimental randomized studies with between- or within-subject parallel designs"

Reply: Ok, we changed the sentence as suggested.

14- Line 138: Capitalise "august"

Reply: Ok, we changed the sentence as suggested (line 135).

15- Methodological quality assessment: For consistency with other studies, the higher the number for the PEDro scale, the better the quality. If future researchers attempt to synthesise studies using the PEDro scale, there would be errors in the interpretation of the evidence. I've just seen your categories have the correct interpretation so making this consistent on line 175 is needed.

Reply: Thank you for your comment, as we noticed an error in the writing which we have corrected (line 172). The classifications are correct and now with the consistency and interpretation clarified.

Results

16- Table 2: Could how the data is being reported in each study be represented by a symbol rather than writing the words?

Reply: We use words to make the text clearer to the reader.

17- Line 200-203: The definition of training status should be presented in the methods section.

Reply: We have changed the sentence for the methods in the "eligibility criteria" section.

18- Line 227: Differences may exist but they may not be significantly different according to null-hypothesis statistical testing. I would add "significantly" before any statement when you are mentioning no difference between groups.

Reply: We use the word "significant" to mention the absence of difference or the presence of difference (lines 216, 220, and 227).

Discussion

19- Line 264: "Influences the development of maximal dynamic and isometric strength"

Reply: Ok, we changed the sentence as suggested (line 258-259).

20- Line 292-293: "Consequently, it is undeniable that heavier load training may be increasingly important..."

Reply: Ok, we changed the sentence as suggested (line 287-288).

21- Line 295: "...changes in agonist-antagonist co-activation..."

Reply: Ok, we changed the sentence as suggested (line 290-291).

22- Line 299-300: Does this sentence mean that high-loads may lead to increases in neural drive compared to low-loads? If so, change to "it seems that high-loads may lead to greater increases in neural drive compared to low-loads..."

Reply: Ok, we changed the sentence as suggested (line 294-295).

23- Line 359-360: I don't think preferential recruitment is the right word choice for this context. The motor units with greater excitability are added in addition to the motor units with lower excitability. So the sentence needs to be made clear that the motor units are added, rather than preferentially recruited which fits Henneman's Size Principle.

Reply: We have changed the sentence taking your comment into consideration.

24- Line 362-363: This sentence doesn't really make sense, but I understand what you're trying to say. This first part of the sentence needs to say that as repetitions are performed, fatigue increases the effort required to produce the same force, which therefore increases motor unit recruitment until the entire motor unit pool is activated and fatigues with the repetitions that are being performed. This is why low-loads are comparable to high-loads, the fatigue-induced increases in effort to overcome the same load requires an increase in motor unit recruitment, which therefore increases mechanical tension, the primary driver of muscular hypertrophy.

Reply: We change according to your suggestion.

25- Line 366: Not only a long time-under-tension but because of the slow movement velocity caused by fatigue, the mechanical tension is actually quite high in magnitude.

Reply: We change according to your suggestion.

26- Line 369-370: The evidence for muscle hypertrophy coming from metabolic stress is mixed at best. The article cited is from 2013 and the evidence has shifted from that point. I would read Henning Wackerhage's review article which discusses metabolic stress as a synergist rather than a direct mechanism (https://pubmed.ncbi.nlm.nih.gov/30335577/).

Reply: Thanks for the suggested reading. Really, Wackerhage's review better supports our writing, and also clarifies the role of metabolic stress in hypertrophy. We have changed the reference for better consistency and change the sentence.

27- Line 373: remove "with low-load". It's implied that you are talking about low-load because you are discussing a minimal load threshold.

Reply: Ok, done.

28- Line 375: "matched volume"

Reply: Ok, done.

29- Line 376: Change "meantime" to "However".

Reply: Ok, Done.

30- Line 380: "...not yet established"

Reply: Ok, done.

31- Line 381-384: From the point above, I agree that low loads should be selected to maximise metabolic stress, but doing this also increases mechanical tension. So you're inducing sufficient levels of mechanical tension either through loading (i.e. high force output) or through fatigue-induced increases in motor unit recruitment and decreases in movement velocity, thus also increasing tension on individual sarcomeres due to the force-velocity relationship. This needs to be made clear in this paragraph because it is an important finding but the muscle physiology/mechanics needs to be applied rather than continuing to use older evidence.

Reply: We agree with your comment. However, there is no really clear evidence in the literature that indicates what is the optimal load percentage (%1RM) for resistance training with low-loads to be able to induce mechanical tension comparable to that performed with high-loads. In fact, there is a very important "load" factor for the mechanical tension to be potentiated. So, we added your comment in a superficial way, without going too deep into the subject.

Conclusions

32- Line 387: "Regarding maximal strength development, the main results of this systematic review..."

Reply: Ok, we changed the sentence as suggested.

33- Line 395: Consistency, healthy adult males or however you would like to phrase it. I'm don't have an affinity for a particular phrasing, as long as it's consistent.

Reply: Ok, we changed the sentence as suggested.

34- Line 408-410: I haven't seen this point discussed in the manuscript until now. I'm not sure how powerful it is to end the article.

Reply: We have removed the sentence in question.

Reviewer 3 Report

Thank you for addressing the comments and suggestions myself and other reviewers made. The new product is significantly different from the draft reviewed initially. Well done. 

Author Response

Reviewer 3

Thank you for addressing the comments and suggestions myself and other reviewers made. The new product is significantly different from the draft reviewed initially. Well done.

Reply: We thank for your revision and we also made a grammatical correction of the text with a native speaker.

Reviewer 4 Report

The paper has been expanded, and deeply rewrote. Now it is suitable for the publication.

Author Response

Reviewer 4

The paper has been expanded, and deeply rewrote. Now it is suitable for the publication.

Reply: We thank for your revision and we also made a grammatical correction of the text with a native speaker.

This manuscript is a resubmission of an earlier submission. The following is a list of the peer review reports and author responses from that submission.

Round 1

Reviewer 1 Report

Major:

This is a systematic review, but there is no pre-registered protocol. What is the reason for that?

The premise of the study is interesting and relevant, and I believe there is merit to this review. Still, the authors must build a stronger case. For example, we have guidelines and studies (even reviews) on the relationship between percentage of load and gains in strength and hypertrophy. The rationale should establish why this is being studied in a stronger manner. Perhaps the authors have found limitations on the guidelines and/or on the research supporting those guidelines. Perhaps they found the guidelines to be biased or incomplete. In particular, the authors should refer to the ACSM's guidelines (try consulting the latest version, i.e., 2021). This will be critical in justifying the relevance of this review. Although the introduction provides some context, I believe the case is not strong enough and requires improvement.

Low, moderate, and high loads: how were these thresholds defined and why? And should we have similar thresholds for exercises involving large muscle masses versus small muscles masses? The literature suggests that the responses are quite different, and even that the number of repetitions at a given % 1RM may change drastically from exercise to exercise. The authors must address this discussion and the implications for their review.

Training to failure: see comments on the rationale before. Not sufficient depth in the arguments provided. Also, studies have suggested that failure may not be required. Please expand on this issue or at least mention that findings are contradictory.

In the last paragraph of the introduction, the authors should reinforce what their review brings anew, and properly refer to the problem of defining these thresholds. Also, according to PRISMA, the goal must contain PICOS elements, but currently it does not. As it is, the goal suggests a much broader review than it is, based on limitations derived from eligibility criteria.

The authors should adjust the structure of the manuscript to better fit the newer PRISMA 2020 guidelines (which were published in March 2021). This will imply that some information will change location, but it will not require changes to the contents of the review.

Eligibility criteria:

  • Why only men? Why this deliberate choice? And this should be widely acknowledged in the abstract, discussion and conclusions.
  • Why limit to participants between 19-44 years of age? Why not 45, or 50? On what basis was this defined? What about 18-years-old? They are adults and are often included.
  • If studies only had moderate loads and high loads, were they still considered? If so, merge intervention and comparators, and explain that.
  • Justify why cross-over or cluster studies were not considered.
  • Were there sensitivity analyses made to understand if randomized and non-randomized studies were comparable? If not, they must be analysed separately in the both the results and the discussion.

Selection criteria: This information is misplaced. Much of this information should be included in the previous section. Here, the selection process should be reported (e.g., who conducted the selection and criteria in case of disagreements).

Search strategy: Where were the terms searched? Title? Title and abstract? Full text? Provide information on this. Also, the “NOT” operator should not have been used and it is strongly advised against by the Cochrane manual. This may exclude articles that are relevant. For example, an article may have had two groups fulfilling PICOS, and an extra group with blood flow restriction. So, the authors may have eliminated articles that could have been included in their study (in the example I provided, two of the three groups could have been considered). Furthermore, since the authors do not state that the keywords were limited to title and abstract, we must assume they could appear in all fields. Thus, even words appearing in the references would have resulted in an article being excluded. As it is, the search strategy cannot be replicated.

Furthermore, given the scope and relevance of this work, and considering that a lot of research is constantly emerging in this topic, the authors should perform an update of searches (from December 2020 onwards) and include novel articles. The search was made 8 months ago, so an update is warranted.

Be consistent - do not report bias in the abstract or manuscript, but methodological quality. The authors did not assess risk of bias, but methodological quality. Although that does not fulfill PRISMA guidelines, the PEDro scale is still currently used, but the terminology must change. The fact that no study was classified >6 should be deeply highlighted in the discussion, conclusion, and abstract.

The results are not exactly surprising, and previous studies have shown this as well. So, the authors should establish a stronger rationale of why they are doing this again, and how their review adds or does something different than the previous literature. I do not mind if the justification consists in merely replicating previous findings, as I believe that replication studies are very much needed. Regardless, this rationale should be explicitly stated by the authors.

83.4% of subjects were untrained. This fact must be deeply highlighted in the abstract and in the discussion. The conclusions must state that results are referring to untrained subjects! Also, they were men by design of the PICOS criteria!

The authors should refer widely to heterogeneity in study design and participants, both in the manuscript and in the abstract.

The results will likely have to be reviewed based on (i) corrections to the search protocol; and (ii) updated searches. This will likely have a strong impact on the discussion.

The discussion will have to more strongly reflect the limitations derived from the PICOS criteria that were established, as well as the evaluations from the PEDro scale. Furthermore, it must prominently highlight the fact that most subjects were untrained. It is not possible to generalize these results to trained subjects (or to women).

Minor:

Both in the title and in the abstract, authors must clarify whether they are talking about absolute load, or relative load. I'm know it is the latter, but this should made explicit.

Specify the databases in the abstract.

In the abstract, provide basic info on sample: mostly men or women? Mostly trained or untrained? Readers should not find out only in the manuscript.

In abstract, highlight the fact that no study scored >6 on PEDro scale.

Training status – when mentioning the classification of Rhea, immediately explain/describe it.

Author Response

Reviewer 1

Major:

1- This is a systematic review, but there is no pre-registered protocol. What is the reason for that?

Reply: It does not have a reason. This paper is part of a doctoral thesis work, where the authors in the course of the work wrote the review as a way to study about the topic, however, the paper has quality, following the PRISMA guidelines and with an important logic in the construction of the writing.

2- The premise of the study is interesting and relevant, and I believe there is merit to this review. Still, the authors must build a stronger case. For example, we have guidelines and studies (even reviews) on the relationship between percentage of load and gains in strength and hypertrophy. The rationale should establish why this is being studied in a stronger manner. Perhaps the authors have found limitations on the guidelines and/or on the research supporting those guidelines. Perhaps they found the guidelines to be biased or incomplete. In particular, the authors should refer to the ACSM's guidelines (try consulting the latest version, i.e., 2021). This will be critical in justifying the relevance of this review. Although the introduction provides some context, I believe the case is not strong enough and requires improvement.

Reply: We significantly changed the introduction as suggested. We tried to reinforce the problematization of our study by showing that in recent articles, the old guidelines may not be the best source of evidence regarding training load. To do this, we explored recent meta-analyses on this topic, as well as a recently published expert position paper (line 66-87; 107-121).

3- Low, moderate, and high loads: how were these thresholds defined and why? And should we have similar thresholds for exercises involving large muscle masses versus small muscles masses? The literature suggests that the responses are quite different, and even that the number of repetitions at a given % 1RM may change drastically from exercise to exercise. The authors must address this discussion and the implications for their review.

Reply: The load thresholds were divided according to NSCA (2016) (Essentials of Strength Training and Conditioning 4th ed: Human kinetics, 2016), and this is set out in the last paragraph of the introduction (line 107-121). Furthermore, the amount of muscle mass involved in a given exercise does not influence the choice of load threshold. For example, Lopez et al. 2021 (Lopez, P., Radaelli, R., Taaffe, D. R., Newton, R. U., Galvão, D. A., Trajano, G. S., Teodoro, J. L., Kraemer, W. J., Häkkinen, K., & Pinto, R. S. (2021). Resistance Training Load Effects on Muscle Hypertrophy and Strength Gain: Systematic Review and Network Meta-analysis. Medicine and science in sports and exercise, 53(6), 1206–1216. https://doi.org/10.1249/MSS.0000000000002585) found no difference in muscle hypertrophy when comparing different load thresholds according to the limb evaluated.

4- Training to failure: see comments on the rationale before. Not sufficient depth in the arguments provided. Also, studies have suggested that failure may not be required. Please expand on this issue or at least mention that findings are contradictory.

Reply: We did not delve into this topic because muscle failure is not the main variable in this systematic review. It is worth noting that when comparing low-loads to high-loads, muscle failure is necessary so that low-loads have the same hypertrophic potential as high-loads (Lasevicius, T., Schoenfeld, B. J., Silva-Batista, C., Barros, T. S., Aihara, A. Y., Brendon, H., Longo, A. R., Tricoli, V., Peres, B. A., & Teixeira, E. L. (2019). Muscle Failure Promotes Greater Muscle Hypertrophy in Low-Load but Not in High-Load Resistance Training. Journal of strength and conditioning research, 10.1519/JSC.0000000000003454).

5- In the last paragraph of the introduction, the authors should reinforce what their review brings anew, and properly refer to the problem of defining these thresholds. Also, according to PRISMA, the goal must contain PICOS elements, but currently it does not. As it is, the goal suggests a much broader review than it is, based on limitations derived from eligibility criteria.

Reply: Considering your comment, we rewrote the last paragraph reinforcing our problematization regarding load thresholds (line 107-121). In addition, we rewrote the objective with the elements of the PICOS strategy (line 25-28; 118-121; 267-270).

6- The authors should adjust the structure of the manuscript to better fit the newer PRISMA 2020 guidelines (which were published in March 2021). This will imply that some information will change location, but it will not require changes to the contents of the review.

Reply: We adjusted the article for the new PRISMA as requested.

Eligibility criteria:

7- Why only men? Why this deliberate choice? And this should be widely acknowledged in the abstract, discussion and conclusions.

Reply: Our choice was made to expand the knowledge of the effects of resistance training performed with different loads in healthy adult males. In addition, we emphasized in the title, abstract, discussion and conclusion our choice (line 2-5; 25-28; 43-46; 118-121;267-270;385-387; 392-396). We limit the results of the systematic review to the population included in the systematic review.

8- Why limit to participants between 19-44 years of age? Why not 45, or 50? On what basis was this defined? What about 18-years-old? They are adults and are often included.

Reply: There was a misunderstanding because the age range selected in the study was 18-40 years. This age range was selected because it has been used in the previous study related to chronic adaptations in resistance training (Benito, P. J., Cupeiro, R., Ramos-Campo, D. J., Alcaraz, P. E., & Rubio-Arias, J. Á. (2020). A Systematic Review with Meta-Analysis of the Effect of Resistance Training on Whole-Body Muscle Growth in Healthy Adult Males. International journal of environmental research and public health, 17(4), 1285. https://doi.org/10.3390/ijerph17041285). Furthermore, after 40 years old, differences in hormonal responses begin to be observed (Harman, S. M., Metter, E. J., Tobin, J. D., Pearson, J., Blackman, M. R., & Baltimore Longitudinal Study of Aging (2001). Longitudinal effects of aging on serum total and free testosterone levels in healthy men. Baltimore Longitudinal Study of Aging. The Journal of clinical endocrinology and metabolism, 86(2), 724–731. https://doi.org/10.1210/jcem.86.2.7219; Labrie, F., Bélanger, A., Luu-The, V., Labrie, C., Simard, J., Cusan, L., Gomez, J. L., & Candas, B. (1998). DHEA and the intracrine formation of androgens and estrogens in peripheral target tissues: its role during aging. Steroids, 63(5-6), 322–328. https://doi.org/10.1016/s0039-128x(98)00007-5).

9- If studies only had moderate loads and high loads, were they still considered? If so, merge intervention and comparators, and explain that.

Reply: For a study to be included it had to have a group with low-loads, as shown in our PICOS strategy (see table 1). For example, two studies (Mangine, G. T., Hoffman, J. R., Gonzalez, A. M., Townsend, J. R., Wells, A. J., Jajtner, A. R., Beyer, K. S., Boone, C. H., Miramonti, A. A., Wang, R., LaMonica, M. B., Fukuda, D. H., Ratamess, N. A., & Stout, J. R. (2015). The effect of training volume and intensity on improvements in muscular strength and size in resistance-trained men. Physiological reports, 3(8), e12472. https://doi.org/10.14814/phy2.12472; Schoenfeld, B. J., Contreras, B., Vigotsky, A. D., & Peterson, M. (2016). Differential Effects of Heavy Versus Moderate Loads on Measures of Strength and Hypertrophy in Resistance-Trained Men. Journal of sports science & medicine, 15(4), 715–722) were excluded because they did not feature low-loads. This can be seen in our study characterization table (see table 3).

10- Justify why cross-over or cluster studies were not considered.

Reply: Cross-over studies were not considered, because in this type of study a washout period is required, and even then, it does not guarantee that the residual effect of the intervention done first will be totally eliminated. On the other hand, cluster type studies were not excluded, because these types of studies are randomized, but the randomization occurs by group of subjects (as opposed to individual subjects). Our PICOS strategy is clear about the type of study included (see table 1).

11- Were there sensitivity analyses made to understand if randomized and non-randomized studies were comparable? If not, they must be analysed separately in the both the results and the discussion.

Reply: A sensitivity analysis is only done in a meta-analysis (we did not perform a meta-analysis). In addition, only two studies did not report randomization, thus we did not differentiate them in our results and discussion.

12- Selection criteria: This information is misplaced. Much of this information should be included in the previous section. Here, the selection process should be reported (e.g., who conducted the selection and criteria in case of disagreements).

Reply: We have included this information in the "eligibility criteria" section as suggested (line 127-135).

13- Search strategy: Where were the terms searched? Title? Title and abstract? Full text? Provide information on this. Also, the “NOT” operator should not have been used and it is strongly advised against by the Cochrane manual. This may exclude articles that are relevant. For example, an article may have had two groups fulfilling PICOS, and an extra group with blood flow restriction. So, the authors may have eliminated articles that could have been included in their study (in the example I provided, two of the three groups could have been considered). Furthermore, since the authors do not state that the keywords were limited to title and abstract, we must assume they could appear in all fields. Thus, even words appearing in the references would have resulted in an article being excluded. As it is, the search strategy cannot be replicated.

Reply: We performed the database search again by excluding the Boolean operator "NOT", as well as restructured the search. We are providing the syntax of the search strategy for each database as supplementary information, so you can see that the search was done considering title and abstract.

14- Furthermore, given the scope and relevance of this work, and considering that a lot of research is constantly emerging in this topic, the authors should perform an update of searches (from December 2020 onwards) and include novel articles. The search was made 8 months ago, so an update is warranted.

Reply: A new search was performed and three studies were included.

15- Be consistent - do not report bias in the abstract or manuscript, but methodological quality. The authors did not assess risk of bias, but methodological quality. Although that does not fulfill PRISMA guidelines, the PEDro scale is still currently used, but the terminology must change. The fact that no study was classified >6 should be deeply highlighted in the discussion, conclusion, and abstract.

Reply: We have changed the terminology and used “assessment of methodological quality" as suggested (line 173-182). In addition, we emphasize the fact that only two studies achieved scores higher than six points for methodological quality (line 36-37; 281-283).

16- The results are not exactly surprising, and previous studies have shown this as well. So, the authors should establish a stronger rationale of why they are doing this again, and how their review adds or does something different than the previous literature. I do not mind if the justification consists in merely replicating previous findings, as I believe that replication studies are very much needed. Regardless, this rationale should be explicitly stated by the authors.

Reply: We rewrote the rationale for the study in the "introduction" topic. Previous systematic reviews and meta-analysis did not consider the moderate-load spectrum in their analysis. In addition, these reviews consider a wide load range as low- (≤60%1RM) and high- (>60%1RM) load. This can be a serious error because, for example, these reviews recommend that muscle hypertrophy may be achieved with low-loads (≤60%1RM), however, Lasevicius, et al. (2018) showed that 20%1RM proved to be a suboptimal load to improve muscle hypertrophy compared to the others. So, this information may be misleading when a wide spectrum of the load is used. We tried to make this clear in the introduction, discussion, and conclusion.

17- 83.4% of subjects were untrained. This fact must be deeply highlighted in the abstract and in the discussion. The conclusions must state that results are referring to untrained subjects! Also, they were men by design of the PICOS criteria!

Reply: We emphasize in the abstract, discussion, and conclusion of the study that our findings were specific to untrained healthy adult males as suggested (line 35; 43-46; 385-387; 392-396).

18- The authors should refer widely to heterogeneity in study design and participants, both in the manuscript and in the abstract.

Reply: We believe that the heterogeneity of the studies and the difference among the participants is already quite clear in the tables where we present the characteristics of the study participants, as well as the characteristics of the included studies. It is implicit from reading the article.

19- The results will likely have to be reviewed based on (i) corrections to the search protocol; and (ii) updated searches. This will likely have a strong impact on the discussion.

Reply: The results were revised according to the methodological adjustments and the new search performed. Taking into account the changes in the results we also revised the discussion and conclusion.

20- The discussion will have to more strongly reflect the limitations derived from the PICOS criteria that were established, as well as the evaluations from the PEDro scale. Furthermore, it must prominently highlight the fact that most subjects were untrained. It is not possible to generalize these results to trained subjects (or to women).

Reply: We try not to extrapolate our results during the discussion and conclusion.

Minor:

21- Both in the title and in the abstract, authors must clarify whether they are talking about absolute load, or relative load. I'm know it is the latter, but this should made explicit.

Reply: We add in the abstract that the study refers to absolute and relative loads (line 35-36).

22- Specify the databases in the abstract.

Reply: Ok, done (line 34).

23- In the abstract, provide basic info on sample: mostly men or women? Mostly trained or untrained? Readers should not find out only in the manuscript.

Reply: All the requested information has been included in the abstract and it has been much improved following your comments.

24- In abstract, highlight the fact that no study scored >6 on PEDro scale.

Reply: Ok, done (line 36-37).

25- Training status – when mentioning the classification of Rhea, immediately explain/describe it.

Ok, done (line 208-211).

Reviewer 1

Major:

1- This is a systematic review, but there is no pre-registered protocol. What is the reason for that?

Reply: It does not have a reason. This paper is part of a doctoral thesis work, where the authors in the course of the work wrote the review as a way to study about the topic, however, the paper has quality, following the PRISMA guidelines and with an important logic in the construction of the writing.

2- The premise of the study is interesting and relevant, and I believe there is merit to this review. Still, the authors must build a stronger case. For example, we have guidelines and studies (even reviews) on the relationship between percentage of load and gains in strength and hypertrophy. The rationale should establish why this is being studied in a stronger manner. Perhaps the authors have found limitations on the guidelines and/or on the research supporting those guidelines. Perhaps they found the guidelines to be biased or incomplete. In particular, the authors should refer to the ACSM's guidelines (try consulting the latest version, i.e., 2021). This will be critical in justifying the relevance of this review. Although the introduction provides some context, I believe the case is not strong enough and requires improvement.

Reply: We significantly changed the introduction as suggested. We tried to reinforce the problematization of our study by showing that in recent articles, the old guidelines may not be the best source of evidence regarding training load. To do this, we explored recent meta-analyses on this topic, as well as a recently published expert position paper (line 66-87; 107-121).

3- Low, moderate, and high loads: how were these thresholds defined and why? And should we have similar thresholds for exercises involving large muscle masses versus small muscles masses? The literature suggests that the responses are quite different, and even that the number of repetitions at a given % 1RM may change drastically from exercise to exercise. The authors must address this discussion and the implications for their review.

Reply: The load thresholds were divided according to NSCA (2016) (Essentials of Strength Training and Conditioning 4th ed: Human kinetics, 2016), and this is set out in the last paragraph of the introduction (line 107-121). Furthermore, the amount of muscle mass involved in a given exercise does not influence the choice of load threshold. For example, Lopez et al. 2021 (Lopez, P., Radaelli, R., Taaffe, D. R., Newton, R. U., Galvão, D. A., Trajano, G. S., Teodoro, J. L., Kraemer, W. J., Häkkinen, K., & Pinto, R. S. (2021). Resistance Training Load Effects on Muscle Hypertrophy and Strength Gain: Systematic Review and Network Meta-analysis. Medicine and science in sports and exercise, 53(6), 1206–1216. https://doi.org/10.1249/MSS.0000000000002585) found no difference in muscle hypertrophy when comparing different load thresholds according to the limb evaluated.

4- Training to failure: see comments on the rationale before. Not sufficient depth in the arguments provided. Also, studies have suggested that failure may not be required. Please expand on this issue or at least mention that findings are contradictory.

Reply: We did not delve into this topic because muscle failure is not the main variable in this systematic review. It is worth noting that when comparing low-loads to high-loads, muscle failure is necessary so that low-loads have the same hypertrophic potential as high-loads (Lasevicius, T., Schoenfeld, B. J., Silva-Batista, C., Barros, T. S., Aihara, A. Y., Brendon, H., Longo, A. R., Tricoli, V., Peres, B. A., & Teixeira, E. L. (2019). Muscle Failure Promotes Greater Muscle Hypertrophy in Low-Load but Not in High-Load Resistance Training. Journal of strength and conditioning research, 10.1519/JSC.0000000000003454).

5- In the last paragraph of the introduction, the authors should reinforce what their review brings anew, and properly refer to the problem of defining these thresholds. Also, according to PRISMA, the goal must contain PICOS elements, but currently it does not. As it is, the goal suggests a much broader review than it is, based on limitations derived from eligibility criteria.

Reply: Considering your comment, we rewrote the last paragraph reinforcing our problematization regarding load thresholds (line 107-121). In addition, we rewrote the objective with the elements of the PICOS strategy (line 25-28; 118-121; 267-270).

6- The authors should adjust the structure of the manuscript to better fit the newer PRISMA 2020 guidelines (which were published in March 2021). This will imply that some information will change location, but it will not require changes to the contents of the review.

Reply: We adjusted the article for the new PRISMA as requested.

Eligibility criteria:

7- Why only men? Why this deliberate choice? And this should be widely acknowledged in the abstract, discussion and conclusions.

Reply: Our choice was made to expand the knowledge of the effects of resistance training performed with different loads in healthy adult males. In addition, we emphasized in the title, abstract, discussion and conclusion our choice (line 2-5; 25-28; 43-46; 118-121;267-270;385-387; 392-396). We limit the results of the systematic review to the population included in the systematic review.

8- Why limit to participants between 19-44 years of age? Why not 45, or 50? On what basis was this defined? What about 18-years-old? They are adults and are often included.

Reply: There was a misunderstanding because the age range selected in the study was 18-40 years. This age range was selected because it has been used in the previous study related to chronic adaptations in resistance training (Benito, P. J., Cupeiro, R., Ramos-Campo, D. J., Alcaraz, P. E., & Rubio-Arias, J. Á. (2020). A Systematic Review with Meta-Analysis of the Effect of Resistance Training on Whole-Body Muscle Growth in Healthy Adult Males. International journal of environmental research and public health, 17(4), 1285. https://doi.org/10.3390/ijerph17041285). Furthermore, after 40 years old, differences in hormonal responses begin to be observed (Harman, S. M., Metter, E. J., Tobin, J. D., Pearson, J., Blackman, M. R., & Baltimore Longitudinal Study of Aging (2001). Longitudinal effects of aging on serum total and free testosterone levels in healthy men. Baltimore Longitudinal Study of Aging. The Journal of clinical endocrinology and metabolism, 86(2), 724–731. https://doi.org/10.1210/jcem.86.2.7219; Labrie, F., Bélanger, A., Luu-The, V., Labrie, C., Simard, J., Cusan, L., Gomez, J. L., & Candas, B. (1998). DHEA and the intracrine formation of androgens and estrogens in peripheral target tissues: its role during aging. Steroids, 63(5-6), 322–328. https://doi.org/10.1016/s0039-128x(98)00007-5).

9- If studies only had moderate loads and high loads, were they still considered? If so, merge intervention and comparators, and explain that.

Reply: For a study to be included it had to have a group with low-loads, as shown in our PICOS strategy (see table 1). For example, two studies (Mangine, G. T., Hoffman, J. R., Gonzalez, A. M., Townsend, J. R., Wells, A. J., Jajtner, A. R., Beyer, K. S., Boone, C. H., Miramonti, A. A., Wang, R., LaMonica, M. B., Fukuda, D. H., Ratamess, N. A., & Stout, J. R. (2015). The effect of training volume and intensity on improvements in muscular strength and size in resistance-trained men. Physiological reports, 3(8), e12472. https://doi.org/10.14814/phy2.12472; Schoenfeld, B. J., Contreras, B., Vigotsky, A. D., & Peterson, M. (2016). Differential Effects of Heavy Versus Moderate Loads on Measures of Strength and Hypertrophy in Resistance-Trained Men. Journal of sports science & medicine, 15(4), 715–722) were excluded because they did not feature low-loads. This can be seen in our study characterization table (see table 3).

10- Justify why cross-over or cluster studies were not considered.

Reply: Cross-over studies were not considered, because in this type of study a washout period is required, and even then, it does not guarantee that the residual effect of the intervention done first will be totally eliminated. On the other hand, cluster type studies were not excluded, because these types of studies are randomized, but the randomization occurs by group of subjects (as opposed to individual subjects). Our PICOS strategy is clear about the type of study included (see table 1).

11- Were there sensitivity analyses made to understand if randomized and non-randomized studies were comparable? If not, they must be analysed separately in the both the results and the discussion.

Reply: A sensitivity analysis is only done in a meta-analysis (we did not perform a meta-analysis). In addition, only two studies did not report randomization, thus we did not differentiate them in our results and discussion.

12- Selection criteria: This information is misplaced. Much of this information should be included in the previous section. Here, the selection process should be reported (e.g., who conducted the selection and criteria in case of disagreements).

Reply: We have included this information in the "eligibility criteria" section as suggested (line 127-135).

13- Search strategy: Where were the terms searched? Title? Title and abstract? Full text? Provide information on this. Also, the “NOT” operator should not have been used and it is strongly advised against by the Cochrane manual. This may exclude articles that are relevant. For example, an article may have had two groups fulfilling PICOS, and an extra group with blood flow restriction. So, the authors may have eliminated articles that could have been included in their study (in the example I provided, two of the three groups could have been considered). Furthermore, since the authors do not state that the keywords were limited to title and abstract, we must assume they could appear in all fields. Thus, even words appearing in the references would have resulted in an article being excluded. As it is, the search strategy cannot be replicated.

Reply: We performed the database search again by excluding the Boolean operator "NOT", as well as restructured the search. We are providing the syntax of the search strategy for each database as supplementary information, so you can see that the search was done considering title and abstract.

14- Furthermore, given the scope and relevance of this work, and considering that a lot of research is constantly emerging in this topic, the authors should perform an update of searches (from December 2020 onwards) and include novel articles. The search was made 8 months ago, so an update is warranted.

Reply: A new search was performed and three studies were included.

15- Be consistent - do not report bias in the abstract or manuscript, but methodological quality. The authors did not assess risk of bias, but methodological quality. Although that does not fulfill PRISMA guidelines, the PEDro scale is still currently used, but the terminology must change. The fact that no study was classified >6 should be deeply highlighted in the discussion, conclusion, and abstract.

Reply: We have changed the terminology and used “assessment of methodological quality" as suggested (line 173-182). In addition, we emphasize the fact that only two studies achieved scores higher than six points for methodological quality (line 36-37; 281-283).

16- The results are not exactly surprising, and previous studies have shown this as well. So, the authors should establish a stronger rationale of why they are doing this again, and how their review adds or does something different than the previous literature. I do not mind if the justification consists in merely replicating previous findings, as I believe that replication studies are very much needed. Regardless, this rationale should be explicitly stated by the authors.

Reply: We rewrote the rationale for the study in the "introduction" topic. Previous systematic reviews and meta-analysis did not consider the moderate-load spectrum in their analysis. In addition, these reviews consider a wide load range as low- (≤60%1RM) and high- (>60%1RM) load. This can be a serious error because, for example, these reviews recommend that muscle hypertrophy may be achieved with low-loads (≤60%1RM), however, Lasevicius, et al. (2018) showed that 20%1RM proved to be a suboptimal load to improve muscle hypertrophy compared to the others. So, this information may be misleading when a wide spectrum of the load is used. We tried to make this clear in the introduction, discussion, and conclusion.

17- 83.4% of subjects were untrained. This fact must be deeply highlighted in the abstract and in the discussion. The conclusions must state that results are referring to untrained subjects! Also, they were men by design of the PICOS criteria!

Reply: We emphasize in the abstract, discussion, and conclusion of the study that our findings were specific to untrained healthy adult males as suggested (line 35; 43-46; 385-387; 392-396).

18- The authors should refer widely to heterogeneity in study design and participants, both in the manuscript and in the abstract.

Reply: We believe that the heterogeneity of the studies and the difference among the participants is already quite clear in the tables where we present the characteristics of the study participants, as well as the characteristics of the included studies. It is implicit from reading the article.

19- The results will likely have to be reviewed based on (i) corrections to the search protocol; and (ii) updated searches. This will likely have a strong impact on the discussion.

Reply: The results were revised according to the methodological adjustments and the new search performed. Taking into account the changes in the results we also revised the discussion and conclusion.

20- The discussion will have to more strongly reflect the limitations derived from the PICOS criteria that were established, as well as the evaluations from the PEDro scale. Furthermore, it must prominently highlight the fact that most subjects were untrained. It is not possible to generalize these results to trained subjects (or to women).

Reply: We try not to extrapolate our results during the discussion and conclusion.

Minor:

21- Both in the title and in the abstract, authors must clarify whether they are talking about absolute load, or relative load. I'm know it is the latter, but this should made explicit.

Reply: We add in the abstract that the study refers to absolute and relative loads (line 35-36).

22- Specify the databases in the abstract.

Reply: Ok, done (line 34).

23- In the abstract, provide basic info on sample: mostly men or women? Mostly trained or untrained? Readers should not find out only in the manuscript.

Reply: All the requested information has been included in the abstract and it has been much improved following your comments.

24- In abstract, highlight the fact that no study scored >6 on PEDro scale.

Reply: Ok, done (line 36-37).

25- Training status – when mentioning the classification of Rhea, immediately explain/describe it.

Ok, done (line 208-211).

Reviewer 2 Report

Thank you for allowing me to read this manuscript on a very important area in the resistance training field of study.  Unfortunately I have chosen to reject based on a large number of flaws in the communication of the background and findings with a large amount of work needed in improving the quality of writing. See below specific comments throughout the manuscript.

Specific comments:

Abstract

Line 22: muscle hypertrophy by definition is the increase in muscle cross-sectional area, so saying "muscle hypertrophy gain" is a redundant phrasing, similar to saying ATM machine. I would revise the sentence to something like the following: "...its effect on the development of muscular strength and muscular hypertrophy" or "...its effect on muscular strength development/gain and muscular hypertrophy".

Line 22-23: The aim reads like an original research study, wouldn't the aim be "to systematically review the literature that compared the effects of resistance training performed with low-, moderate- and high-loads..."

Line 23: similar type comment to the above, add maximal strength development because hypertrophy is saying that there is a change in size but saying "maximal strength" does not imply change.

Line 28: "skeletal muscle fiber", need to be slightly more specific here. Are you referring to number, size, MHC isoform analysis, or all of the above? I'm thinking that if you are referring to all of them, I think adding "analysis" would suffice.

Introduction

Line 53: Be specific to which guidelines you are talking about. Saying "main" appears subjective or influenced by opinion. I would change to say where the actual guidelines have come from.

Line 61: muscle hypertrophy gain stated here, same comment as above

Line 63: "...has shown to be an important factor"

Line 67: remove "a"

Line 66-69: Not sure that the Schoenfeld study is the best article to use here to state your claim surrounding motor unit recruitment with low loads, purely on the basis that it was a multi-joint exercise. To accurately assess muscle activation and different loads, an isolated exercise model would need to be used to remove the impacts of technique breakdown as a reason why the set was ceased. 

Line 69: The peak sEMG amplitude does not imply maximal activation of motor units, it only gives muscle activation which is a function of recruitment and firing rate. You could get a higher amplitude purely from a greater firing rate. Some more background research may need to be done in this paragraph to strengthen the claim surrounding how low loads are regulated by the neuromuscular system. See: "Greater Electromyographic Responses Do Not Imply Greater Motor Unit Recruitment and ‘Hypertrophic Potential’ Cannot Be Inferred" to start which has great citations surrounding recruitment and EMG amplitude.

Line 71: "a probable benefit"

Line 72-74: This sentence is inaccurate. The use of low-loads is appropriate for muscular strength and hypertrophy, particualrly hypertrophy, as when they are performed to muscular failure, there is a high degree of effort or "central motor drive" etc. which then activates a greater amount of motor units to continue to overcome the resistance. Time-under-tension in isolation does not lead to greater recruitment, magnitude of effort does which can be achieved with high loads or low loads to failure. Consider revising this paragraph with some more background information on how EMG amplitude is being interpreted in this context and also adding in how low-load resistance training works to increase particular markers (which needs to be specific).

Line 97: "...which leaves a lack of well-defined ranges..."

Line 103: re-word to state that this manuscript is a review AND that the outcomes that you mention clearly state that you are looking at the development (or similar) of strength and hypertrophy.

Methods:

Table 1: Generally really like how this was done! Some comments below:

Participants: The inclusion of only men is puzzling as there is no rationale for it in the introduction or anywhere in the manuscript, likewise with age. I'm not sure if this is critical to the research question? Particularly when training status is not defined as strictly.

Intervention: Remove "sessions", intervention implies there are sessions involved

Comparators: Remove "sessions"

Outcomes: Remove "in the result outcomes"

Study design: "randomized or non-randomized"

Information sources

Line 122: What is meant by a "highly sensitive search"

If the last search was done in December 2020, I would do the search again because I think this area is quite a popular one in the resistance training field of study.

Study Selection

Line 143: change "send" to "sent"

Results

Study Characteristics

Line 183: consider changing the sentence to: "To classify training status, we implented the model proposed by Rhea [43].

Table 2: If available, the standard deviation of all studies should be presented in age, height, and body mass. Also the mean and SD and range rows at the end are not necessary and won't be accurate as it is likely that you don't have every data point from each study. I would consider removing unless you do have every data point from each study pertaining to these variables.

Line 190 and 192: replace "strength in 1RM" with "strength using a 1RM".

Line 190 and 191: consider having these two sentences combined into one to state: "Regarding the maximal strength assessment, 16 studies assessed dynamic strength using a 1RM (76.2%) [10-14, 16-18, 24, 25, 33, 37-39, 41, 42] and another six studies assessed isometric strength (28.6%) [13, 34, 35, 37, 191 38, 40]."

Line 193: replace "in" with "using a" MVIC

Line 194: I don't understand the context of the following fragment "and three studies assessed dynamic strength in 1RM". Only because in line 190 you state that 16 studies measure dynamic strength using a 1RM.

Line 196: "assessed muscle hypertrophy using ultrasound" is a much better way to phrase this result. This same logic should be applied for the muscular strength section.

Line 201: "most studies", I would state exactly how many studies demonstrated a significant improvement to make reading this section a bit easier

Line 214-215: change "mainly the repetitions are performed to failure" to "on the provision that repetitions in each set are performed to muscular failure".

The results section don't appear to present a commentary on the programs that were prescribed and quite critically, the loading that was used. Loading is the primary research question of this piece so the results will need a critical commentary of the loadings and how they were described to give the reader a better understanding of the current literature. Something I was interested in was seeing how high and low loads were used when studies measured strength isometrically.

Table 3 is difficult to read and likely needs to be in landscape format if the journal formatting guidelines permit.

Discussion

Considering my concerns in the previous sections, I believe the discussion will change significantly so I have chosen not to read the discussion in great detail. The discussion I thought was a literature review in itself which rarely related back to the findings which is a concern in a systematic review. I like how the discussion was structured, particularly for muscular hypertrophy where the commentary was situated with high loads and low loads with emphasis on muscular failure, followed by a paragraph on loads that are too low to exert any changes in muscle size.

Unfortunately, for now I recommend to reject this article for now as I think the writing of this article is in need of serious improvement regarding background knowledge and presentation of results and data from other studies.

Author Response

Reviewer 2

Thank you for allowing me to read this manuscript on a very important area in the resistance training field of study.  Unfortunately, I have chosen to reject based on a large number of flaws in the communication of the background and findings with a large amount of work needed in improving the quality of writing. See below specific comments throughout the manuscript.

Reply: Dear reviewer, the study has been significantly reformulated and improved its quality considerably, and we believe that it will add a lot to the practical field and help coaches to prescribe training more assertively and safely.

Specific comments:

Abstract

1- Line 22: muscle hypertrophy by definition is the increase in muscle cross-sectional area, so saying "muscle hypertrophy gain" is a redundant phrasing, similar to saying ATM machine. I would revise the sentence to something like the following: "...its effect on the development of muscular strength and muscular hypertrophy" or "...its effect on muscular strength development/gain and muscular hypertrophy".

Reply: We changed as suggested (line 24-25).

2- Line 22-23: The aim reads like an original research study, wouldn't the aim be "to systematically review the literature that compared the effects of resistance training performed with low-, moderate- and high-loads..."

Reply: The objective has been changed as suggested. In addition, we have incorporated as PRISMA suggests aspects of the PICOS strategy to make the objective more specific (line 25-28).

3- Line 23: similar type comment to the above, add maximal strength development because hypertrophy is saying that there is a change in size but saying "maximal strength" does not imply change.

Reply: We changed as suggested (line 27).

4- Line 28: "skeletal muscle fiber", need to be slightly more specific here. Are you referring to number, size, MHC isoform analysis, or all of the above? I'm thinking that if you are referring to all of them, I think adding "analysis" would suffice.

Reply: This part of the abstract has been reformulated and the term "skeletal muscle fiber" has been removed (line 32).

Introduction

5- Line 53: Be specific to which guidelines you are talking about. Saying "main" appears subjective or influenced by opinion. I would change to say where the actual guidelines have come from.

Reply: We changed as suggested (line 60).

6- Line 61: muscle hypertrophy gain stated here, same comment as above

Reply: From the comments of the other reviewers this paragraph has been reformulated and the location you indicated has been removed. Therefore, please re-evaluate the paragraph and consider our changes (line 66-87).

7- Line 63: "...has shown to be an important factor"

Reply: From the comments of the other reviewers, this paragraph has been reformulated and the location you indicated has been removed. Therefore, please re-evaluate the paragraph and consider our changes (line 66-87).

8- Line 67: remove "a"

Reply: From the comments of the other reviewers, this paragraph has been reworded and the location you indicated has been removed. Therefore, please re-evaluate the paragraph and consider our changes (line 66-87).

9- Line 66-69: Not sure that the Schoenfeld study is the best article to use here to state your claim surrounding motor unit recruitment with low loads, purely on the basis that it was a multi-joint exercise. To accurately assess muscle activation and different loads, an isolated exercise model would need to be used to remove the impacts of technique breakdown as a reason why the set was ceased.

Reply: From the comments of the other reviewers, this paragraph has been reformulated and the location you indicated has been removed. Therefore, please re-evaluate the paragraph and consider our changes (line 66-87).

10- Line 69: The peak sEMG amplitude does not imply maximal activation of motor units, it only gives muscle activation which is a function of recruitment and firing rate. You could get a higher amplitude purely from a greater firing rate. Some more background research may need to be done in this paragraph to strengthen the claim surrounding how low loads are regulated by the neuromuscular system. See: "Greater Electromyographic Responses Do Not Imply Greater Motor Unit Recruitment and ‘Hypertrophic Potential’ Cannot Be Inferred" to start which has great citations surrounding recruitment and EMG amplitude.

Reply: From the comments of the other reviewers, this paragraph has been reformulated and the location you indicated has been removed. Therefore, please re-evaluate the paragraph and consider our changes (line 66-87).

11- Line 71: "a probable benefit"

Reply: From the comments of the other reviewers, this paragraph has been reformulated and the passage you indicated has been removed. Therefore, please re-evaluate the paragraph and consider our changes (line 66-87).

12- Line 72-74: This sentence is inaccurate. The use of low-loads is appropriate for muscular strength and hypertrophy, particularly hypertrophy, as when they are performed to muscular failure, there is a high degree of effort or "central motor drive" etc. which then activates a greater amount of motor units to continue to overcome the resistance. Time-under-tension in isolation does not lead to greater recruitment, magnitude of effort does which can be achieved with high loads or low loads to failure. Consider revising this paragraph with some more background information on how EMG amplitude is being interpreted in this context and also adding in how low-load resistance training works to increase particular markers (which needs to be specific).

Reply: From the comments of the other reviewers, this paragraph has been reformulated and the passage you indicated has been removed. Therefore, please re-evaluate the paragraph and consider our changes (line 66-87).

13- Line 97: "...which leaves a lack of well-defined ranges..."

Reply: From the comments of the other reviewers, this paragraph has been reformulated and the sentence that you indicated has been removed. Therefore, please re-evaluate the paragraph and consider our changes (line 107-121).

14- Line 103: re-word to state that this manuscript is a review AND that the outcomes that you mention clearly state that you are looking at the development (or similar) of strength and hypertrophy.

Reply: The objective has been reformulated after your comment above and those of the other reviewers. We add aspects of the PICOS strategy into the objective as suggested by PRISMA to make the objective more specific (line 118-121).

Methods:

15- Table 1: Generally, really like how this was done! Some comments below:

Reply: We appreciate your comments below.

16- Participants: The inclusion of only men is puzzling as there is no rationale for it in the introduction or anywhere in the manuscript, likewise with age. I'm not sure if this is critical to the research question? Particularly when training status is not defined as strictly.

Reply: Our choice of males and delimiting age was in line with another systematic review (Benito, P. J., Cupeiro, R., Ramos-Campo, D. J., Alcaraz, P. E., & Rubio-Arias, J. Á. (2020). A Systematic Review with Meta-Analysis of the Effect of Resistance Training on Whole-Body Muscle Growth in Healthy Adult Males. International journal of environmental research and public health, 17(4), 1285. https://doi.org/10.3390/ijerph17041285).We made our choice to expand the knowledge about the effects of training load on chronic adaptations in healthy adult males. In addition, we emphasized our choice in the title, abstract, discussion, and conclusion. It is worth noting, that the age previously reported in the article (i.e., 19-44 years) was wrong, as it was previously defined that we would select articles where the participants were 18-40 years old.

17- Intervention: Remove "sessions", intervention implies there are sessions involved

Reply: Ok, done (see table 1).

18- Comparators: Remove "sessions"

Reply: Ok, done (see table 1).

19- Outcomes: Remove "in the result outcomes"

Reply: Ok, done (see table 1).

20- Study design: "randomized or non-randomized"

Reply: Ok, done (see table 1).

Information sources

21- Line 122: What is meant by a "highly sensitive search"

Reply: This term was removed because we detected gaps in the search strategy after suggestions from the other reviewers. Therefore, we reformulated the search strategy, performed the searches again, and adapted the entire methodology and flow diagram for PRISMA version 2020 (see methods and figure 1).

22- If the last search was done in December 2020, I would do the search again because I think this area is quite a popular one in the resistance training field of study.

Reply: We appreciate your comment, and after your suggestion, we performed the database search again on August 22, 2021.

Study Selection

23- Line 143: change "send" to "sent"

Reply: Ok, done (line 162).

Results

Study Characteristics

24- Line 183: consider changing the sentence to: "To classify training status, we implanted the model proposed by Rhea [43].

Reply: We modified this sentence and described what was considered as untrained, recreationally trained, and highly trained (Rhea MR. Determining the magnitude of treatment effects in strength training research through the use of the effect size. J Strength Cond Res. 2004 Nov;18(4):918-20. doi: 10.1519/14403.1. PMID: 15574101.), as suggested by other reviewers. I ask you to reconsider the sentence and make your considerations (line 208-211).

25- Table 2: If available, the standard deviation of all studies should be presented in age, height, and body mass. Also, the mean and SD and range rows at the end are not necessary and won't be accurate as it is likely that you don't have every data point from each study. I would consider removing unless you do have every data point from each study pertaining to these variables.

Reply: Excellent comment, we provided, when possible, the SD or SE for all studies on all variables cited (i.e., body mass, age, and height). Also, as suggested, we excluded the mean, SD, and the range provided in the final rows of the table 2.

26- Line 190 and 192: replace "strength in 1RM" with "strength using a 1RM".

Reply: Ok, done (line 218 and 221).

27- Line 190 and 191: consider having these two sentences combined into one to state: "Regarding the maximal strength assessment, 16 studies assessed dynamic strength using a 1RM (76.2%) [10-14, 16-18, 24, 25, 33, 37-39, 41, 42] and another six studies assessed isometric strength (28.6%) [13, 34, 35, 37, 191 38, 40]."

Reply: Ok, done.  We rephrased this part (line 230-239).

28- Line 193: replace "in" with "using a" MVIC

Reply: Ok, done (line 221).

29- Line 194: I don't understand the context of the following fragment "and three studies assessed dynamic strength in 1RM". Only because in line 190 you state that 16 studies measure dynamic strength using a 1RM.

Reply: Ok, done.  We fix this mistake (line 221-224).

30- Line 196: "assessed muscle hypertrophy using ultrasound" is a much better way to phrase this result. This same logic should be applied for the muscular strength section.

Reply: Ok, done.  We rephrased this part (line 218-221).

31- Line 201: "most studies", I would state exactly how many studies demonstrated a significant improvement to make reading this section a bit easier.

Reply: Ok, done (line 218).

32- Line 214-215: change "mainly the repetitions are performed to failure" to "on the provision that repetitions in each set are performed to muscular failure".

Reply: Ok, we changed the sentence as suggested (line 243-244).

33- The results section don't appear to present a commentary on the programs that were prescribed and quite critically, the loading that was used. Loading is the primary research question of this piece so the results will need a critical commentary of the loadings and how they were described to give the reader a better understanding of the current literature. Something I was interested in was seeing how high and low loads were used when studies measured strength isometrically.

Reply: The main objective of the results section is to perform a descriptive analysis of the main results of the studies included in the systematic review. The form in which the loads were prescribed in each study is found in the table (see table 3). Therefore, in order not to repeat the information contained in the tables, we chose not to perform a descriptive analysis of them. The analysis of the training programs and their respective applied loads, as well as their results on the investigation variables, are in the discussion (line 269-398).

34- Table 3 is difficult to read and likely needs to be in landscape format if the journal formatting guidelines permit.

Reply: We changed the table to landscape format as suggested (see table 3).

Discussion

Considering my concerns in the previous sections, I believe the discussion will change significantly so I have chosen not to read the discussion in great detail. The discussion I thought was a literature review in itself which rarely related back to the findings which is a concern in a systematic review. I like how the discussion was structured, particularly for muscular hypertrophy where the commentary was situated with high loads and low loads with emphasis on muscular failure, followed by a paragraph on loads that are too low to exert any changes in muscle size.

Unfortunately, for now I recommend to reject this article for now as I think the writing of this article is in need of serious improvement regarding background knowledge and presentation of results and data from other studies.

Reviewer 3 Report

General Comment:

Well written manuscript. I do not notice any fundamental errors in this manuscript. What follows are suggestions for consideration regarding the text.

Abstract

  • Well done. Perhaps a bit wordy, but each point is made independently and provides succinct information.

Introduction

  • L46 – consider removing the word ‘several’. The implication of specificity invites the need to provide said specific information.
  • L75 – consider rewording this sentence
  • L76 - muscle is misspelled
  • L95 – consider using a different word than ‘optimized’. Many in the strength training world do not believe such a level can be achieved. Specific goal oriented programming is a more realistic take on the idea.
  • L103 – change was to were

Methods and Materials

  • L107 – a systematic review was performed. The use of ‘We’ shifts to focus of the information to the authors from the methods of the research.
  • L110 – Reword this sentence. Grammatical errors.
  • L143 – send should be changed to sent
  • L144 – reword the last sentence. Some has no contextual meaning here.
  • L159 – A risk of bias assessment…

Results

  • L170 – A flow diagram….
  • L179 – selected for systematic review. Remove the.
  • L182 – Poorly written sentence. Please reword to focus on the procedure.
  • L201 – remove ‘the’. Regarding maximal….
  • L222 – Table S1? Where is this table?

Discussion

  • L249-273 – there seems to be a number of very extended sentences. Please consider re-writing these.
  • L278 – Perhaps Therefore should begin a new sentence.
  • L315 – Results of this research….

Conclusion

  • L368 and L372 – Additionally and addition are use quite close to one another. Perhaps a different word choice for one would allow the dialogue to flow better.
  • L372 – reword the sentence – grammatical issues.

References

  • All citations begin with the lead author’s last name, followed by their initials. Then all other authors are reported with initials first then last name. I found this odd.

Author Response

Reviewer 3

General Comment:

Well written manuscript. I do not notice any fundamental errors in this manuscript. What follows are suggestions for consideration regarding the text.

Reply: We thank you for your comment and answer all your questions below.

Abstract

1- Well done. Perhaps a bit wordy, but each point is made independently and provides succinct information.

Reply: Despite your comment other reviewers have requested changes. Therefore, we have made some changes and tried to keep the summary informative and succinct (line 22-47).

Introduction

2- L46 – consider removing the word ‘several’. The implication of specificity invites the need to provide said specific information.

Reply: The word was removed as suggested (line 53).

3- L75 – consider rewording this sentence

Reply: we rewrite the sentence as suggested (line 88-89).

4- L76 - muscle is misspelled

Ok, we corrected the writing of the word (line 88).

5- L95 – consider using a different word than ‘optimized’. Many in the strength training world do not believe such a level can be achieved. Specific goal oriented programming is a more realistic take on the idea.

Reply: We changed "optimized" to "efficient” (line 109).

6- L103 – change was to were

Reply: The objective of the study has changed in writing, I ask you to read it and evaluate again if the use of “was” is correct? (line 118-121).

Methods and Materials

7- L107 – a systematic review was performed. The use of ‘We’ shifts to focus of the information to the authors from the methods of the research.

Reply: We rewrite the sentence (line 123-125)

8- L110 – Reword this sentence. Grammatical errors.

Reply: We rewrite the sentence (line 127-135).

9- L143 – send should be changed to sent

Reply: Ok, done (line 162).

10- L144 – reword the last sentence. Some has no contextual meaning here.

Reply: This sentence was removed.

11- L159 – A risk of bias assessment…

Reply: This topic has been reworded as suggested by the other reviewers, please re-evaluate again (line 173-182).

Results

12- L170 – A flow diagram

Reply: Ok, done (line 193).

13- L179 – selected for systematic review. Remove the.

Reply: Ok, done (line 204-205).

14- L182 – Poorly written sentence. Please reword to focus on the procedure.

Reply: Ok, done.  We rephrased this part (line 209-212).

15- L201 – remove ‘the’. Regarding maximal….

Reply: Ok, done (line 228).

16- L222 – Table S1? Where is this table?

Reply: It was submitted as supplementary material, but we will remove it from the supplementary material after this peer review process.

Discussion

17- L249-273 – there seems to be a number of very extended sentences. Please consider re-writing these.

Reply: We modified as suggested (line 284-308).

18- L278 – Perhaps Therefore should begin a new sentence.

Reply: Ok, done (315).

19- L315 – Results of this research….

Reply: Ok, done (line 358).

Conclusion

20- L368 and L372 – Additionally and addition are use quite close to one another. Perhaps a different word choice for one would allow the dialogue to flow better.

21- L372 – reword the sentence – grammatical issues.

Reply: We reword the sentence (line 418-424).

References

21- All citations begin with the lead author’s last name, followed by their initials. Then all other authors are reported with initials first then last name. I found this odd.

Reply: This has been corrected.

Reviewer 4 Report

The study suffer from a generalization bias. In high performance sports, loads to increase maximal strength (1 RM) are quite different. For example, a top level weight lifter, fo have an improvement of 5% can need one year training with loads ranging from 80 to 100% of 1RPM. Thus it must be considered the sport and the starting level of the athletes. From a methodological point of view, it does not make a sense a generalization on 500 athletes..

Introduction:

"Resistance training is the main exercise intervention used to enhance maximal strength and muscle hypertrophy"

This is not true, because maximal strength can be increased with plyometric training as well, and this is the method of election in certain sports (for example, jumpings in track and field).

"Thus, the aims of the present study was to compare the effect of resistance training per-103 formed with low-, moderate- and high-loads on maximal strength and muscle hypertrophy" ; here must be stated: in healthy non trained subjects.

The range 19-44 years of the subjects introduce significant biases.  Thus this is a study for non trained population.

line 212: Moreover, comparisons among  groups (i.e., low-, moderate- and high-load) did not show significant difference in the cross-sectional area or muscle thickness.

This statment does not have a sense, because all exercises were performed at failure, so the load is identical irrespective of the weight. All along the paper there is a confusion between training load and weigth used. 

Conclusions" lne 355

"The main results of this systematic review indicated that regarding maximal strength, studies suggest that high-loads promote greater improvements compared to  low-loads.

This is well known.

line 361: "For muscle hypertrophy, studies indicate that a wide spectrum of load (i.e., 30 to 90%  1RM) may be used for untrained or trained individuals.

Must be specified for untrained because mostly of the studie considered are for untrained. Don't mix the 2 groups. Also must be specified : at failure.

The title must report: untrained healthy individuals

Author Response

Reviewer 4

1- The study suffer from a generalization bias. In high performance sports, loads to increase maximal strength (1 RM) are quite different. For example, a top level weightlifter, for have an improvement of 5% can need one year training with loads ranging from 80 to 100% of 1RM. Thus, it must be considered the sport and the starting level of the athletes. From a methodological point of view, it does not make a sense a generalization on 500 athletes.

Reply: Our study after the reviews included a total of 563 participants, of which 454 were untrained and 109 recreationally trained in resistance training. Therefore, the generalization was taking into account that more than 80% of the participants were untrained. Thus, our results are important for this population (untrained and perhaps recreationally trained), as emphasized in our conclusion.

Introduction:

2- "Resistance training is the main exercise intervention used to enhance maximal strength and muscle hypertrophy"

This is not true, because maximal strength can be increased with plyometric training as well, and this is the method of election in certain sports (for example, jumpings in track and field).

Reply: We have removed the “main” word to avoid erroneous interpretations (line 52-53).

4- "Thus, the aims of the present study was to compare the effect of resistance training per-103 formed with low-, moderate- and high-loads on maximal strength and muscle hypertrophy”; here must be stated: in healthy non trained subjects.

Reply: The objective has been reformulated and as suggested we have added your suggestion (line 25-28; 120-123; 270-273).

5- The range 19-44 years of the subjects introduce significant biases. Thus this is a study for non trained population.

Reply: There was a misunderstanding because the age range selected in the study was 18-40 years. This age range was selected because it has been used in the previous study related to chronic adaptations in resistance training (Benito, P. J., Cupeiro, R., Ramos-Campo, D. J., Alcaraz, P. E., & Rubio-Arias, J. Á. (2020). A Systematic Review with Meta-Analysis of the Effect of Resistance Training on Whole-Body Muscle Growth in Healthy Adult Males. International journal of environmental research and public health, 17(4), 1285. https://doi.org/10.3390/ijerph17041285). Furthermore, after 40 years old, differences in hormonal responses begin to be observed (Harman, S. M., Metter, E. J., Tobin, J. D., Pearson, J., Blackman, M. R., & Baltimore Longitudinal Study of Aging (2001). Longitudinal effects of aging on serum total and free testosterone levels in healthy men. Baltimore Longitudinal Study of Aging. The Journal of clinical endocrinology and metabolism, 86(2), 724–731. https://doi.org/10.1210/jcem.86.2.7219; Labrie, F., Bélanger, A., Luu-The, V., Labrie, C., Simard, J., Cusan, L., Gomez, J. L., & Candas, B. (1998). DHEA and the intracrine formation of androgens and estrogens in peripheral target tissues: its role during aging. Steroids, 63(5-6), 322–328. https://doi.org/10.1016/s0039-128x(98)00007-5).

6- line 212: Moreover, comparisons among groups (i.e., low-, moderate- and high-load) did not show significant difference in the cross-sectional area or muscle thickness.

This statment does not have a sense, because all exercises were performed at failure, so the load is identical irrespective of the weight. All along the paper there is a confusion between training load and weigth used.

Reply: We report whether there was a significant difference between comparing the different loads (i.e., low, moderate, and high) and included it in the studies. Also, nowhere in the manuscript do we use the term "weight", as all resistance training guidelines and studies on the topic use the term "load". Therefore, in Author’s opinion, there was no confusion regarding terms, because we standardized this in the article.

Conclusions" line 355

7- "The main results of this systematic review indicated that regarding maximal strength, studies suggest that high-loads promote greater improvements compared to low-loads.

This is well known.

Reply:  Yes. But even so it is part of the results because it reinforces the other reviews.

8- line 361: "For muscle hypertrophy, studies indicate that a wide spectrum of load (i.e., 30 to 90%1RM) may be used for untrained or trained individuals.

Must be specified for untrained because mostly of the studies considered are for untrained. Don't mix the 2 groups. Also, must be specified: at failure.

Reply: We make this point strongly here (line 408-412).

9- The title must report: untrained healthy individuals

Reply: The title has been reworded and as suggested we have inserted your suggestion (line 2-5).
